# DFRot: Achieving Outlier-Free and Massive Activation-Free for Rotated LLMs with Refined Rotation

## Abstract

Rotating the activation and weight matrices to reduce the influence of outliers in large language models (LLMs) has recently attracted significant attention, particularly in the context of model quantization. Prior studies have shown that in low-precision quantization scenarios, such as 4-bit weights and 4-bit activations (W4A4), randomized Hadamard transforms can achieve significantly higher accuracy than randomized orthogonal transforms. Notably, the reason behind this phenomena remains unknown. In this paper, we find that these transformations show substantial improvement in eliminating outliers for common tokens and achieve similar quantization error. The primary reason for the accuracy difference lies in the fact that randomized Hadamard transforms can slightly reduce the quantization error for tokens with massive activations while randomized orthogonal transforms increase the quantization error. Due to the extreme rarity of these tokens and their critical impact on model accuracy, we consider this a long-tail optimization problem, and therefore construct a simple yet effective method: a weighted loss function. Additionally, we propose an optimization strategy for the rotation matrix that involves alternating optimization of quantization parameters while employing orthogonal Procrustes transforms to refine the rotation matrix. This makes the distribution of the rotated activation values more conducive to quantization, especially for tokens with massive activations. Our method enhances the Rotated LLMs by achieving dual free, *Outlier-Free* and *Massive Activation-Free*, dubbed as DFRot. Extensive experiments demonstrate the effectiveness and efficiency of DFRot. By tuning the rotation matrix using just a single sample, DFRot achieves a perplexity improvement of 0.25 and 0.21 on W4A4KV4 and W4A4KV16, respectively, for LLaMA3-8B, a model known for its quantization challenges. Code is anonymously available at https://anonymous.4open.science/r/DFRot-8FE3.

## 1 Introduction

Large Language Models (LLMs) have shown exceptional abilities across numerous domains. Cutting-edge open-source models like LLaMA (Touvron et al., 2023) and Mistral (Jiang et al., 2023), along with proprietary LLMs such as GPT (Achiam et al., 2023) and Gemini (Team et al., 2023), are now being applied in a wide range of applications, including natural language understanding (Zellers et al., 2019; Hendrycks et al., 2020), machine translation (Zhang et al., 2023), content generation (Mo et al., 2024), and recommendation systems (Wu et al., 2023).

However, the remarkable success of LLMs is largely reliant on significant computational resources. LLMs often consist of billions of parameters, making them not only resource-intensive to train but also challenging to deploy on devices with limited computational capacity, such as mobile phones and edge devices. Additionally, the high memory and processing demands not only drive up hardware costs but also significantly increase energy consumption, leading to serious deployment concerns. To address these challenges, researchers and engineers are actively exploring various model compression techniques (Frantar et al., 2022; Xiao et al., 2023; Lin et al., 2024a; Yao et al., 2022; Frantar & Alistarh, 2023; Ashkboos et al., 2024a). These techniques aim to reduce the size of LLMs while maintaining their performance as effectively as possible, achieving a balance between

efficiency and accuracy. Among the various methods, Post-Training Quantization (PTQ) provides a training-free approach, or one with minimal training cost for calibration purposes Nagel et al. (2019); Li et al. (2021), allowing for rapid and efficient quantization. Compared to Quantization-Aware Training (QAT), which requires multiple rounds of fine-tuning, PTQ incurs significantly lower computational costs. This makes it an appealing option for quantizing LLMs.

Unfortunately, the presence of outliers in the activations (Dettmers et al., 2022; Zeng et al., 2022) often leads to a significant reduction in model accuracy when PTQ is applied directly. To address this problem, earlier approaches have either scaled weights and activations (Xiao et al., 2023; Wei et al., 2023; Shao et al., 2023), shifting the quantization challenges from activations to weights, or employed mixed-precision techniques to isolate outliers (Dettmers et al., 2022), thereby minimizing the LLM's quantization error.

Recent research (Ashkboos et al., 2024b) has demonstrated that rotating activations in LLMs can effectively eliminate most outliers while preserving computational invariance, ensuring that the LLM's output remains identical to its original results. Moreover, the rotation matrices can be merged into the weights, imposing no additional burden on network inference. This innovative computational invariance (Ashkboos et al., 2024a) has garnered significant attention from researchers.

Although rotation is widely recognized as an important method for the quantization of LLMs, there remain many unresolved issues. For example, as shown in Table 1, when activations are reduced to 4 bits, the reasons why randomized Hadamard transforms (RH) often achieve significant improvement compared to randomized orthogonal transforms (RO) (Ashkboos et al., 2024b; Liu et al., 2024) have not yet been fully understood. However, while directly training rotation matrices can yield good results (Liu et al., 2024), the training process will cause substantial computational resources and adds complexity to the quantization process.

In this paper, we first investigate the underlying reasons why RH outperforms RO. We find that for ordinary tokens consisting primarily of outliers (Achiam et al., 2023), both RO and RH transformations can equally reduce quantization error when applied to these tokens. In contrast, for special tokens with *massive activations* (Sun et al., 2024), using RO on these activations surprisingly leads to an increase in quantization error. Our experiments show that this inability to efficiently manage massive activations greatly restricts the accuracy of quantized LLMs. On the other hand, while RH performs better than RO, it only manages to maintain or slightly reduce the quantization error for these large activations. This observation indicates that both transformation methods struggle to effectively manage massive activations in LLM quantization.

Building on these insights, we propose a novel optimization method to enhance the performance of quantized LLMs, achieving both *Outlier-Free* and *Massive Activation-Free*, *e.g.* dual free (DFRot). By treating scarce tokens with massive activations as long-tail distributed data, we develop a simple yet effective weighted loss function. Additionally, we introduce an alternating optimization approach to refine the rotation matrices and quantization parameters, further minimizing quantization error. Extensive experiments demonstrate the effectiveness of our proposed method. Specifically, by tuning the rotation matrix with just a single sample and additional 8 minutes, DFRot achieves a PPL improvement of 0.25 and 0.21 on W4A4KV4 and W4A4KV16 for LLaMA3-8B, a model recognized for its quantization challenges (Huang et al., 2024).

## 2 RELATED WORK

Reducing quantization error is essential for model quantization. However, as reported by LLM.int8() (Dettmers et al., 2022), simply quantizing LLM to INT8 results in significant accuracy degradation due to the presence of outliers. To handle emerging outliers, LLM.int8() introduces a mixed-precision decomposition scheme. Although it can preserve the model's accuracy, the complexity of fine-grained decomposition always leads to computational overhead and potential performance bottlenecks. Currently, research in LLM quantization predominantly focuses on eliminating outliers through scale invariance and rotation invariance.

### 2.1 ELIMINATING OUTLIERS VIA SCALE INVARIANCE

The initial idea behind suppressing outliers through scale invariance stems from the observation that weights are easier to quantize than activations, and outliers in activations often appear in a few fixed

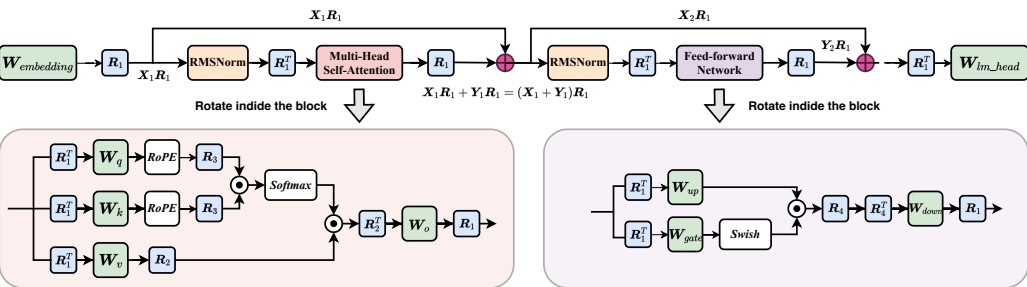

Figure 1: An illustration of rotational invariance in the LLaMA architecture. The rotation matrix $R_1$ can be integrated into the residual connection, ensuring the network retains rotational invariance. The rotation inner the block can further reducing outliers in the block. Both of them make LLM fewer outliers and be easier to quantize. The rotation matrix $R_1$, $R_1^T$, $R_2$, $R_2^T$ and $R_4^T$ can be integrated with the adjunct weights. $R_3$ and $R_4$ need to compute online.

channels Dettmers et al., 2022. Based on this, SmoothQuant (Xiao et al., 2023) first proposes that we can offline migrate the quantization difficulty from activations to weights via scale invariance. SmoothQuant enables an INT8 quantization of both weights and activations for all the matrix multiplications in LLMs. Furthermore, Outlier Suppression+ (Wei et al., 2023) proposes a fast and stable scheme to effectively calculate scaling values, achieving a better balance in quantization burden. To reduce manual design and further enhance quantization performance in extremely low-bit quantization, OmniQuant (Shao et al., 2023) introduces Learnable Weight Clipping and Learnable Equivalent Transformation, efficiently optimizing the quantization process for both weight-only and weight-activation quantization. In the clipping W4A8 quantization, QQQ (Zhang et al., 2024) proposes to dynamically handle outliers through adaptive smoothing. QServe (Lin et al., 2024b) proposes SmoothAttention to effectively mitigate the accuracy degradation caused by 4-bit KV quantization. Both QQQ and QServe have effectively enhanced the performance of LLMs in W4A8 quantization.

## 2.2 ELIMINATING OUTLIERS VIA ROTATION INVARIANCE

Although scale invariance can reduce outliers and improve quantization performance, it merely transfers the outliers from activations to weights and has not eliminated them fundamentally. When the magnitude of the outliers is large, scaling struggles to achieve an effective balance between weights and activations. Recently, researchers have found that applying rotation matrices to networks can effectively reduce outliers without increasing the complexity of LLMs. QuIP Chee et al. (2024) is the first to suggest that quantization can benefit from the incoherence between weight and Hessian matrices. It employed randomized orthogonal matrices generated by Kronecker product to enhance their incoherence. QuIP# (Tseng et al., 2024) replaces the randomized orthogonal matrices with randomized Hadamard matrices, which are faster and possess better theoretical properties. QuaRot (Ashkboos et al., 2024b) is the first work to apply rotational invariance (Ashkboos et al., 2024a) for model quantization. QuaRot finds that randomized Hadamard transformations yield better results compared to randomized orthogonal transformations. SpinQuant (Liu et al., 2024) further extends the rotation matrices to a trainable space and applied Cayley optimization (Li et al., 2020) to refine them, achieving significant improvements across diverse datasets.

## 3 METHODOLOGY

### 3.1 PRELIMINARY

To remove outliers in the input activations $X_1$, a rotation matrix $R_1$ is applied to the input matrix $X^1$, resulting in a new input activation $X_1 R_1$. $R_1$ satisfies $R_1 R_1^T = R_1^T R_1 = I$ and $|R_1| = 1$. Using the LLaMA architecture as an example, $X_1 R_1$ is then passed to the RMSNorm, which satisfies the commutation property: $\text{RMSNorm}(X_1 R_1) = \text{RMSNorm}(X_1) R_1$ (Ashkboos et al., 2024a). Here, we assume that RMSNorm operates on each row $i$ of the activations $X_1$ as $X_{1,i} \leftarrow X_{1,i}/|X_{1,i}|$. This commutation property implies that multiplying the input of RMSNorm by $R_1$ is equivalent to multiplying the RMSNorm output by $R_1$ as well.

Table 1: WikiText-2 perplexity ($\downarrow$) results for RO and RH for LLaMA and Mistral models. The 4-4-4, 4-4-16, 4-8-16 represent W4A4KV4, W4A4KV16, W4A8KV16 respectively. We show the failed GPTQ using NaN and the perplexity results>100 by Inf. QuaRot.FP16() denotes retaining tokens with massive activations as FP16.

| Method | LLaMA2-7B | | | LLaMA2-13B | | | LLaMA3-8B | | | Mistral-7B-v0.3 | | |
|---|---|---|---|---|---|---|---|---|---|---|---|---|
| | 4-4-4 | 4-4-16 | 4-8-16 | 4-4-4 | 4-4-16 | 4-8-16 | 4-4-4 | 4-4-16 | 4-8-16 | 4-4-4 | 4-4-16 | 4-8-16 |
| GPTQ | NaN | NaN | NaN | Inf | Inf | 6.01 | Inf | Inf | 7.29 | Inf | Inf | 8.39 |
| (RO) QuaRot | 7.96 | 7.71 | 5.61 | 6.00 | 5.92 | 4.99 | 10.54 | 10.15 | 6.52 | 6.05 | 5.98 | 5.40 |
| (RO) QuaRot.FP16() | **6.17** | **6.10** | - | **5.38** | **5.34** | - | **7.83** | **7.68** | - | **5.79** | **5.73** | - |
| (RH) QuaRot | 6.27 | 6.20 | 5.61 | 5.51 | 5.46 | 5.01 | 8.20 | 8.02 | 6.52 | 5.81 | 5.75 | 5.40 |
| (RH) QuaRot.FP16() | **6.17** | **6.10** | - | **5.40** | **5.37** | - | **7.82** | **7.67** | - | **5.78** | **5.73** | - |

The output of LayerNorm is then passed into the subsequent linear blocks. With the introduction of $R_1$, the input to these linear layers is altered. To ensure that the output from the linear layers remains unchanged, $R_1^T$ is multiplied by the weight matrix $W$, resulting in a new weight matrix $R_1^T W$, which can be calculated offline. Since $R_1^T R_1 = I$, the output from the linear layer remains unaffected. This *computational invariance* property of LLMs ensure the introduction of the rotation matrices without changing the original results.

A similar approach can be applied to rest layers within an LLM block. As shown in Figure 1, by transforming the weight matrices in the Multi-Head Attention (MHA) as $R_1^T W_q$, $R_1^T W_k$, $R_1^T W_v$, and $W_o R_1$, and the weights in the Feed-Forward Network (FFN) as $R_1^T W_{up}$, $R_1^T W_{gate}$, and $W_{down} R_1$, the hidden features within both MHA and FFN remain unchanged. Consequently, the output feature $Y_1$ is transformed into $Y_1 R_1$, which will sum with the residual input $X_1 R_1$ satisfies $X_1 R_1 + Y_1 R_1 = (X_1 + Y_1) R_1 = X_2 R_1$. The output will serve as the input for the next LLM block. Similarly, by transforming $W_{lm\_head}$ to $R_1^T W_{lm\_head}$, the network output will remain unchanged.

Moreover, we can introduce additional rotation matrices to further mitigate outliers between layers. As illustrated in Figure 1, head-wise rotation matrices $R_2$ and $R_2^T$ can be applied to $W_v$ and $W_o$, while $R_3$ can be inserted for **Query** and **Key** after RoPE. Additionally, $R_4$ and $R_4^T$ can be placed between the Swish activation and $W_{down}$. These strategies help further suppress outliers and reduce quantization error without affecting the block's output. In this paper, we focus exclusively on $R_1$. For $R_2$, $R_3$, and $R_4$, we adopt the settings from QuaRot (Ashkboos et al., 2024b) by setting them to random Hadamard matrices.

## 3.2 WHY THE RANDOMIZED HADAMARD IS BETTER?

Based on the computational invariance described in Section 3.1, it is evident that the choice of rotation matrices is critical for ensuring the accuracy performance of the quantized model. Therefore, a natural question arises: *What type of rotation matrix offers the most advantageous properties?* We begin by focusing on RO and RH, as both QuaRot (Ashkboos et al., 2024b) and SpinQuant (Liu et al., 2024) have shown that the latter delivers substantial improvements over the former in LLMs. We conducted experiments by applying RO and RH to the LLaMA and Mistral models, followed by weight quantization using GPTQ under various settings. The results are shown in Table 1, benefiting from the outlier elimination through rotation, we find that for 8-bit activation quantization, both RO and RH lead to significant performance improvements compared to standard quantization. Additionally, no substantial difference is observed between the two methods. However, **under 4-bit token-wise activation quantization, RH significantly outperforms RO**.

To investigate the performance differences between RH and RO under 4-bit activation setting, we plot the corresponding quantization error after applying 4-bit quantization to the multiple tokens. We also display the quantization error for the baseline setting where quantization is applied without rotating the activation to better understand the impact of using the rotation matrix. As shown in Figure 2, compared to the no rotation (NR), both RO and RH effectively reduce the quantization error for most tokens across different models. While RH slightly lowers the quantization error, the difference between the two methods is minimal for the majority of tokens. This leads to the

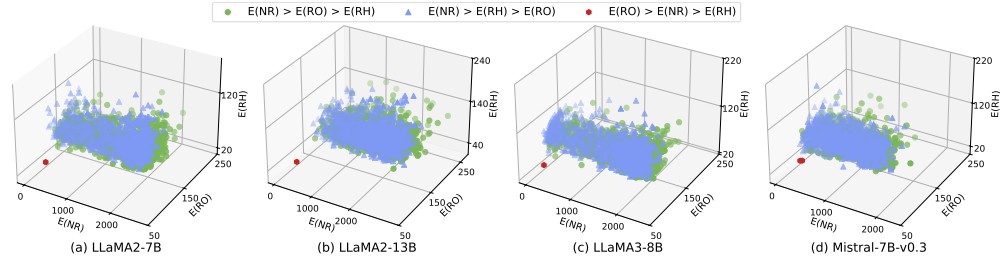

Figure 2: Comparison of 4-bit activation quantization error E(·) for each token with NR, RO and RH for (a) LLaMA2-7B, (b) LLaMA-2-13B, (c) LLaMA3-8B and (d) Mistral-7B-v0.3. The tokens are from model.layers.6.post_attention_layernorm. Best viewed in color.

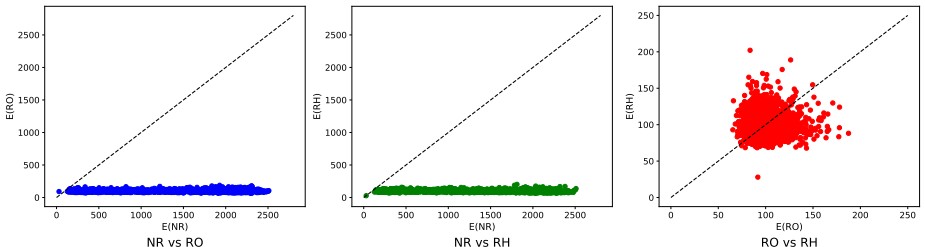

Figure 3: Comparison of 2D 4-bit quantization errors for tokens with NR, RO and RH for LLaMA3-8B from Figure 2.

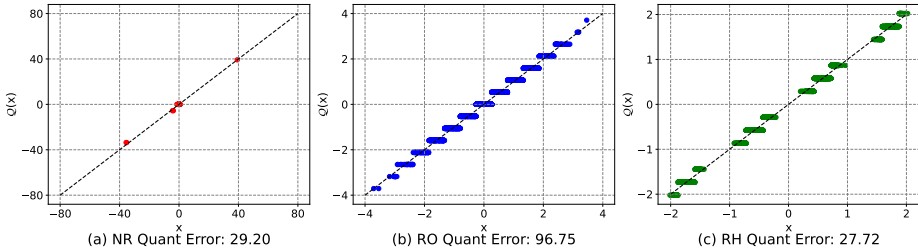

Figure 4: Comparison of 4-bit quantization error for the token with massive activation with NR, RO and RH for LLaMA3-8B from Figure 2.

question: **What explains the significant difference in accuracy during quantization when their quantization errors are so similar?**

To answer this question, we turn our attention to massive activation (Sun et al., 2024), a rare but significant feature in LLMs. Since each token has a fixed $L_2$ norm after RMSNorm processing, tokens with massive activation naturally exhibit smaller quantization errors when quantized to 4-bit. As shown in Figure 2, the red points represent tokens with massive activation. While most tokens show large quantization errors under NR, these special tokens display significantly smaller errors, which can be observed from Figure 3. Figure 4 presents the quantization error distribution for tokens with massive activation after applying RO, RH, and NR. Surprisingly, the rotation operations do not significantly reduce quantization errors for these tokens. In fact, compared to NR, RO greatly increases their quantization error, while RH only marginally reduces it. This leads us to question whether tokens with massive activation are the primary cause of the significant accuracy discrepancies between RH and RO.

To investigate this further, we build upon QuaRot by retaining tokens with massive activations in FP16 format for both RO and RH, while applying 4-bit quantization to the remaining input tokens. Therefore, we can conclude that the fundamental reason for the performance disparity between RO and RH is that **RH more effectively reduces the quantization error for tokens with massive activations in 4-bit activation quantization**.

### 3.3 OPTIMIZATION OBJECTIVES AND CALIBRATION DATA SELECTION

The evaluation results in Section 3.2 show that applying 4-bit quantization to activations leads to significant quantization errors due to the large volume of activations, ultimately causing accuracy

degradation. While encoding these activations more precisely could alleviate the issue, it results in a mixed quantization approach that is not well-suited for current GPU platforms. A good rotation matrix $\boldsymbol{R}_1$ should minimize the different between the original input $\boldsymbol{x}$ and its quantized version, namely:

$$\mathcal{L}(\boldsymbol{R}_1, \boldsymbol{g}) = \mathbb{E}_{\boldsymbol{x}}\left[\|\boldsymbol{x}\boldsymbol{R}_1 - \mathcal{Q}_{\boldsymbol{g}}(\boldsymbol{x}\boldsymbol{R}_1)\|_2^2\right], \tag{1}$$

where $\boldsymbol{x} \in \mathcal{R}^C$ is the token vector from a calibration dataset $\boldsymbol{X}^{cal}$, $C$ is the number of channels. $\boldsymbol{R}_1$ satisfies $\boldsymbol{R}_1 \boldsymbol{R}_1^T = \boldsymbol{I}$, $\boldsymbol{g}$ is the quantization parameters and $\mathcal{Q}_{\boldsymbol{g}}(\boldsymbol{x})$ is the quantization representation of the $\boldsymbol{x}$. The size of $\mathcal{Q}_g(\boldsymbol{x})$ is the same to the $\boldsymbol{x}$. The expectation $\mathbb{E}[\cdot]$ is taken over the token distribution. For simplicity in analysis, we utilize the mean squared error, denoted as $\|\cdot\|_2$.

To better adapt $\boldsymbol{R}_1$ to the massive activations, we adjust it by optimizing the following loss function:

$$\mathcal{L}(\boldsymbol{R}_1, \boldsymbol{g}) = \mathbb{E}_{\boldsymbol{x} \in \boldsymbol{X}^{cal} \setminus \boldsymbol{X}^m}\left[\|\boldsymbol{x}\boldsymbol{R}_1 - \mathcal{Q}_{\boldsymbol{g}}(\boldsymbol{x}\boldsymbol{R}_1)\|_2^2\right] + \gamma \mathbb{E}_{\boldsymbol{x} \in \boldsymbol{X}^m}\left[\|\boldsymbol{x}\boldsymbol{R}_1 - \mathcal{Q}_{\boldsymbol{g}}(\boldsymbol{x}\boldsymbol{R}_1)\|_2^2\right]. \tag{2}$$

where $\boldsymbol{X}^m \subseteq \boldsymbol{X}^{cal}$ denotes the subset of tokens with massive activations, while $\boldsymbol{X}^{cal} \setminus \boldsymbol{X}^m$ represents the remaining tokens. During calibration, we apply a weighted loss to prioritize the quantization error on tokens with massive activations, with $\gamma$ representing the weight.

The motivation behind this principle stems from the observations in Table 1. Since $\boldsymbol{X}^m$ is the key factor contributing to the performance gap between RO and RH. Simply optimizing $\boldsymbol{R}_1$ over the entire $\boldsymbol{X}^{cal}$ fails to specifically target $\boldsymbol{X}^m$. Additionally, compared to the NR approach in Table 1, RO also significantly improves performance, indicating that reducing the outliers on $\boldsymbol{X}^{cal} \setminus \boldsymbol{X}^m$ can enhance the performance of the quantization method. However, optimizing only for $\boldsymbol{X}^m$ risks overfitting, which could increase the quantization error for $\boldsymbol{X} \setminus \boldsymbol{X}^m$, ultimately degrading the model's overall performance. Hence, it is crucial to optimize both $\boldsymbol{X}^m$ and $\boldsymbol{X} \setminus \boldsymbol{X}^m$. Using a weighted approach to optimize the quantization loss is a straightforward yet highly effective method. Ablation studies in Section 4.2 further demonstrate the advantages of this strategy.

## 3.4 SOLUTION METHODS

Optimizing $\boldsymbol{R}_1$ is a challenging task. Since $\boldsymbol{R}_1$ influences every MHA and FFN in the network, adjusting the activation distribution in one layer impacts the quantization outcomes across all layers. This makes it difficult to optimize layer by layer or block by block (Shao et al., 2023; Wei et al., 2023). A straightforward approach is to use training methods for quantization-aware fine-tuning of the rotation matrix across the entire network (Liu et al., 2024). However, this approach necessitates fine-tuning the entire network. Although it does not require retaining the gradients of the weights or the corresponding states in the optimizer, it still demands substantial computational resources during the quantization process.

In this paper, we focus on improving the effectiveness of rotation matrices in mitigating outliers in activation values. Intuitively, we hypothesize that a rotation matrix that minimizes quantization error will lead to fewer activation outliers and, consequently, better performance. Drawing inspiration from Simsiam (Chen & He, 2021), we propose to regard quantization representation $\mathcal{Q}_{\boldsymbol{g}}(\boldsymbol{x}\boldsymbol{R_1})$ as cluster centroids $\boldsymbol{\eta_x}$. In the context, optimizing $\boldsymbol{R}_1$ and $\boldsymbol{g}$ is equivalent to optimizing $\boldsymbol{R}_1$ and $\boldsymbol{\eta}$, which can be viewed as an implementation of an Expectation-Maximization (EM)-like algorithm, as shown in the following equation:

$$\min_{\boldsymbol{R}_1, \boldsymbol{\eta}} \mathcal{L}(\boldsymbol{R}_1, \boldsymbol{\eta}) = \mathbb{E}_{\boldsymbol{x} \in \boldsymbol{X}^{cal} \setminus \boldsymbol{X}^m}\left[\|\boldsymbol{x}\boldsymbol{R}_1 - \boldsymbol{\eta_x}\|_2^2\right] + \gamma \mathbb{E}_{\boldsymbol{x} \in \boldsymbol{X}^{cal}}\left[\|\boldsymbol{x}\boldsymbol{R}_1 - \boldsymbol{\eta_x}\|_2^2\right], \tag{3}$$

where $\boldsymbol{\eta_x} = \mathcal{Q}_{\boldsymbol{g}}(\boldsymbol{x}\boldsymbol{R}_1)$. This formulation is analogous to k-means clustering (Macqueen, 1967), and $\boldsymbol{R}_1$ acts like the kernel function, representing the learnable rotation matrix. Similar to k-means clustering, the problem described in Eq 3 can be approached using an alternating algorithm, where one set of variables is fixed while solving for the other. Formally, we can alternate between solving these two subproblems:

$$\boldsymbol{\eta}^t \leftarrow \arg\min_{\boldsymbol{\eta}} \mathcal{L}\left(\boldsymbol{R}_1^{t-1}, \boldsymbol{\eta}\right) \tag{4}$$

$$\boldsymbol{R}_1^t \leftarrow \arg\min_{\boldsymbol{R}_1} \mathcal{L}\left(\boldsymbol{R}_1, \boldsymbol{\eta}^t\right) \tag{5}$$

where $t$ represents the iteration index of the alternating rounds, and $\boldsymbol{\eta}^t$ and $\boldsymbol{R}_1^t$ denote the values of $\boldsymbol{\eta}$ and $\boldsymbol{R}_1$ at round $t$.

**Solving for the cluster centroids $\eta_x$**   The set of quantization parameters $g\{s, z\}$ further contains the quantization scale $s$ and zero point $z$. Assume we apply the static quantization, the $s^t, z^t$ and $\eta_x$ can be solved by the following equations:

$$s^t, z^t \leftarrow \arg\min_{s,z} \mathbb{E}_x \left[ \left\| xR_1^{t-1} - \mathcal{Q}_{s,z}(xR_1^{t-1}) \right\|_2^2 \right], \eta_x^t \leftarrow \mathcal{Q}_{s^t, z^t}(xR_1^{t-1}) \tag{6}$$

In the case of dynamic asymmetric per-token quantization, we can independently determine the optimal quantization scheme for solving $s_x$ and $z_x$ for each $xR_1$:

$$\eta_x = \mathcal{Q}_g(xR_1) = \text{clamp}\left( \left\lfloor \frac{xR_1}{s} \right\rceil + z, 0, 2^N - 1 \right),$$

$$\text{where } s_x = \frac{\alpha \max(xR_1) - \beta \min(xR_1)}{2^N - 1}, z_x = -\left\lfloor \frac{\beta \min(xR_1)}{s_x} \right\rceil \tag{7}$$

where $\lfloor \cdot \rceil$ indicates round operation, $N$ is the bitwidth, and $\alpha$ and $\beta$ is the clip ratio for upper bound and lower bound of quantization, respectively.

**Solving for $R_1$.**   Eq 5 is well-known as Procrustes problem (Mulaik, 2009). which involves finding the optimal rotation matrix $R_1$ that best aligns two sets of points, minimizing the Frobenius norm of their difference. The solution to this problem can be obtained through Singular Value Decomposition (SVD). Specifically, given input matrices $X = \{x\}$ and its quantized version $\mathcal{Q}_g(X) = \{\mathcal{Q}_g(x)\}$, the optimal $R_1$ can be found:

$$R_1 = UV^T, \text{where } U, \Sigma, V^T = \text{SVD}(X^T \mathcal{Q}_{g^t}(X)). \tag{8}$$

where we treat the quantization parameters $g^t$ as a constant.

**One-step optimization.**   To find an improved rotation matrix $R_1$ and quantization parameters $g$, we perform the iterative process shown in Eq 4 and Eq 5 with just 100 rounds, which already yields significantly better performance, as demonstrated in the evaluation (Section 4). Specifically, a calibration set $X^{cal}$ is randomly sampled from $X$, the iterative process can be specified as:

$$s^t, z^t \leftarrow \arg\min_{s,z} \sum_{x \in X^{cal}} \left[ \left\| xR_1^{t-1} - \mathcal{Q}_{s,z}(xR_1^{t-1})) \right\|_2^2 \right], \eta_x^t \leftarrow \mathcal{Q}_{s^t, z^t}(xR_1^{t-1}), \tag{9}$$

then the resulting quantization parameters will be used to produce the rotation matrix:

$$R_1^t \leftarrow \arg\min_{R_1} \sum_{x \in X^{cal}} \left[ \left\| xR_1 - \eta_x^t \right\|_2^2 \right] \tag{10}$$

The detailed algorithm is provided in Algorithm 1 in Appendix.

## 4  EXPERIMENTS

**Experiment settings.**   We implemented DFRot based on QuaRot[1]. In this paper, to simplify the problem, we apply dynamic asymmetric per-token quantization for activation values without searching for clip ratios, and we fix $(\alpha, \beta)$ to $(1.0, 1.0)$. The KV-cache is quantized using asymmetric quantization with a group size of 128 and a constant clipping ratio of 1.0. RTN and GPTQ (Frantar et al., 2022) are used for weight with per-channel symmetric quantization, where a linear search for the clipping ratio is applied to minimize squared error. We use 128 samples from the WikiText-2 (Merity et al., 2016) training set, each with a sequence length of 2048, as the calibration dataset for GPTQ quantization. We use a RH to initialize the rotation matrix and optimize it for 100 iterations.

### 4.1  MAIN RESULTS

**Language Generation Task.**   Firstly, we evaluate DFRot on a language generation task and compare it with QuaRot. We quantize the weights using both the RTN and GPTQ methods. Table 2 shows the perplexity of LLaMA and Mistral models. As shown, compared to QuaRot, DFRot

---

[1]https://github.com/spcl/QuaRot

Table 2: WikiText-2 perplexity (↓) results for LLaMA and Mistral. The 4-4-4 and 4-4-16 represent W4A4KV4, W4A4KV16, respectively. We show the failed GPTQ experiments using NaN and the perplexity results>100 by Inf.

| Method | LLaMA2-7B | | LLaMA2-13B | | LLaMA3-8B | | Mistral-7B-v0.3 | |
|---|---|---|---|---|---|---|---|---|
| Baseline | 5.47 | | 4.88 | | 6.14 | | 5.32 | |
| Extra Time | +8min | | +20min | | +8min | | +8min | |
| | 4-4-4 | 4-4-16 | 4-4-4 | 4-4-16 | 4-4-4 | 4-4-16 | 4-4-4 | 4-4-16 |
| RTN | NaN | NaN | Inf | Inf | Inf | Inf | Inf | Inf |
| QuaRot-RTN | 9.04 | 8.69 | 6.31 | 6.23 | 11.06 | 10.47 | 6.38 | 6.29 |
| DFRot-RTN | 7.68 | 7.47 | 6.21 | 6.12 | 9.67 | 9.35 | 6.36 | 6.27 |
| GPTQ | NaN | NaN | Inf | Inf | Inf | Inf | Inf | Inf |
| QuaRot-GPTQ | 6.27 | 6.20 | 5.51 | 5.47 | 8.20 | 8.02 | 5.81 | 5.75 |
| DFRot-GPTQ | 6.21 | 6.14 | 5.47 | 5.39 | 7.95 | 7.81 | 5.81 | 5.76 |

Table 3: Zero-shot accuracy (↑) of LLaMA and Mistral with GPTQ on PIQA (PQ), Wino-Grande (WG), HellaSwag (HS), Arc-Easy (A-e), Arc-Challenge (A-c), and LAMBADA (LA).

| Model | Method | W-A-KV | PQ | WG | HS | A-e | A-c | LA | Avg. |
|---|---|---|---|---|---|---|---|---|---|
| LLaMA2-7B | FP16 | 16-16-16 | 79.11 | 68.98 | 75.99 | 74.54 | 46.42 | 73.88 | 69.82 |
| | QuaRot | 4-4-16 | 76.06 | 65.67 | 73.00 | 69.82 | 42.24 | 69.42 | 66.03 |
| | | 4-4-4 | 76.33 | 64.96 | 72.69 | 68.60 | 41.64 | 68.58 | 65.47 |
| | DFRot | 4-4-16 | 77.15 | 65.82 | 73.17 | 69.78 | 44.37 | 70.66 | 66.83 |
| | | 4-4-4 | 76.22 | 64.96 | 72.41 | 70.75 | 42.66 | 69.92 | 66.15 |
| LLaMA2-13B | FP16 | 16-16-16 | 80.52 | 72.22 | 79.39 | 77.48 | 49.15 | 76.75 | 72.58 |
| | QuaRot | 4-4-16 | 77.91 | 68.51 | 75.94 | 73.57 | 46.25 | 72.97 | 69.19 |
| | | 4-4-4 | 78.73 | 70.40 | 75.82 | 73.74 | 46.33 | 72.73 | 69.63 |
| | DFRot | 4-4-16 | 78.73 | 69.30 | 76.99 | 72.69 | 45.82 | 75.41 | 69.82 |
| | | 4-4-4 | 79.82 | 68.43 | 76.70 | 72.64 | 46.59 | 75.33 | 69.92 |
| LLaMA3-8B | FP16 | 16-16-16 | 80.79 | 72.85 | 79.16 | 77.78 | 53.33 | 76.03 | 73.32 |
| | QuaRot | 4-4-16 | 74.92 | 66.61 | 73.39 | 70.29 | 44.54 | 67.71 | 66.24 |
| | | 4-4-4 | 75.14 | 66.54 | 72.32 | 68.64 | 42.41 | 66.04 | 65.18 |
| | DFRot | 4-4-16 | 76.22 | 68.03 | 73.92 | 70.41 | 45.65 | 68.87 | 67.18 |
| | | 4-4-4 | 75.68 | 66.77 | 73.56 | 70.29 | 45.14 | 68.99 | 66.74 |
| Mistral-7B-v0.3 | FP16 | 16-16-16 | 82.26 | 73.88 | 80.41 | 78.20 | 52.30 | 75.32 | 73.73 |
| | QuaRot | 4-4-16 | 79.54 | 69.30 | 77.81 | 75.51 | 47.95 | 73.76 | 70.65 |
| | | 4-4-4 | 79.38 | 69.06 | 77.36 | 74.54 | 48.29 | 73.55 | 70.36 |
| | DFRot | 4-4-16 | 79.87 | 69.53 | 78.24 | 75.88 | 48.46 | 73.01 | 70.83 |
| | | 4-4-4 | 80.36 | 69.61 | 78.01 | 75.55 | 47.95 | 72.39 | 70.65 |

achieves improvements in most cases. Notably, DFRot achieves the most significant improvement on the LLaMA3-8B model with W4A4KV4 and W4A4KV16 using GPTQ, outperforming QuaRot by 0.25 and 0.21, respectively. Similar to QuaRot, DFRot does not require any retraining process and only needs an additional sample to optimize the rotation matrix. On a single NVIDIA A100 GPU, optimizing the rotation matrix takes an extra 8 minutes for embeddings of 4096 (LLaMA2-7B, LLaMA3-8B & Mistral-7B-v0.3) and 20 minutes for 5120 (LLaMA2-13B), resulting in minimal overhead. It demonstrates that DFRot has wide applicability and can serve as a cost-effective method to enhance the quantization performance of rotated LLMs.

**Zero-Shot Tasks.** Following QuaRot, we also evaluate DFRot on the following six important zero-shot tasks: PIQA (Bisk et al., 2020), WinoGrande (Sakaguchi et al., 2021), HellaSwag (Zellers et al., 2019), Arc (Easy and Challenge) (Clark et al., 2018) and LAMBADA (Radford et al., 2019). We used lm_eval==0.4.3 (Gao et al., 2024) and GPTQ for our experiments, with default parameters and weight quantization, respectively. Table 3 shows the accuracy of DFRot on the above tasks as well as the average score. As can be seen, DFRot consistently achieves improvements compared to

QuaRot across all tasks. For example, DFRot achieves a 1.56% accuracy improvement compared to QuaRot on the LLaMA3-8B model with W4A4KV4 quantization settings.

## 4.2 ABLATION STUDIES

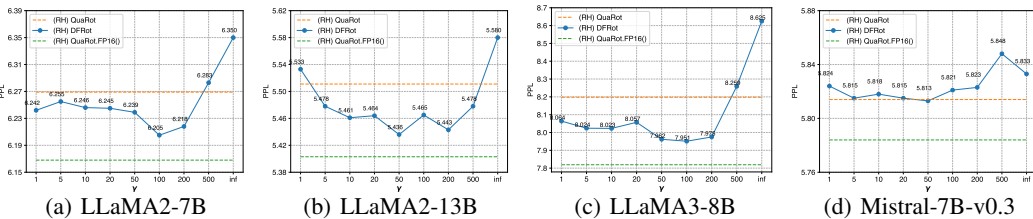

(a) LLaMA2-7B  (b) LLaMA2-13B  (c) LLaMA3-8B  (d) Mistral-7B-v0.3

Figure 5:  (RH) Comparison of WikiText-2 perplexity results under different $\gamma$ for W4A4KV4. Weight is quantized via GPTQ. $\gamma \to \infty$ denotes we only optimize quantization error for $\boldsymbol{X}^m$.

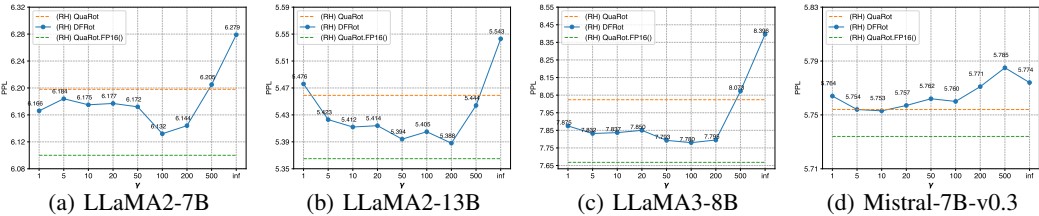

(a) LLaMA2-7B  (b) LLaMA2-13B  (c) LLaMA3-8B  (d) Mistral-7B-v0.3

Figure 6:  (RH) Comparison of WikiText-2 perplexity results under different $\gamma$ for W4A4KV16. Weight is quantized via GPTQ. $\gamma \to \infty$ denotes we only optimize quantization error for $\boldsymbol{X}^m$.

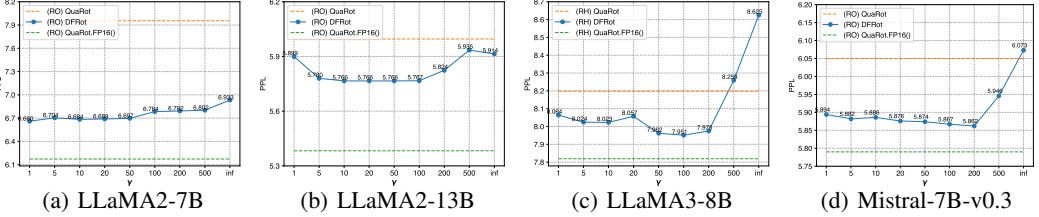

(a) LLaMA2-7B  (b) LLaMA2-13B  (c) LLaMA3-8B  (d) Mistral-7B-v0.3

Figure 7:  (RO) Comparison of WikiText-2 perplexity results under different $\gamma$ for W4A4KV4. Weight is quantized via GPTQ. $\gamma \to \infty$ denotes we only optimize quantization error for $\boldsymbol{X}^m$.

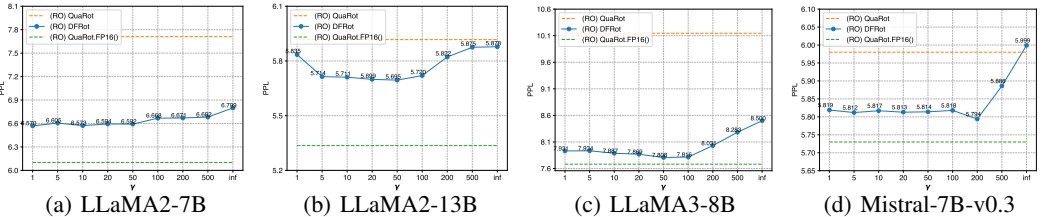

(a) LLaMA2-7B  (b) LLaMA2-13B  (c) LLaMA3-8B  (d) Mistral-7B-v0.3

Figure 8:  (RO) Comparison of WikiText-2 perplexity results under different $\gamma$ for W4A4KV16. Weight is quantized via GPTQ. $\gamma \to \infty$ denotes we only optimize quantization error for $\boldsymbol{X}^m$.

**Choice of $\gamma$.** To further understand the effect of hyperparameters in DFRot, we conducted an ablation study on Wikitext-2 PPL to investigate the impact of different $\gamma$ settings for W4A4KV4 and W4A4KV16. As seen in Figures 5 and 6, when $\gamma$ ranges between 50 and 200, DFRot achieves significant improvements across various LLaMA models using $\mathrm{RH}$. Notably, on the LLaMA3-8B model, known for its quantization challenges, we observed a PPL improvement of over 0.2. If we set $\gamma = 1$ and treat $\boldsymbol{X}^m$ and $\boldsymbol{X} \setminus \boldsymbol{X}^m$ equally to minimize their quantization errors, it may reduce the quantization loss of $\boldsymbol{X} \setminus \boldsymbol{X}^m$ but increase the quantization loss of $\boldsymbol{X}^m$, ultimately resulting in a performance decline on the LLaMA2-13B. Conversely, if we set $\gamma \to \infty$ and only optimize the quantization error for $\boldsymbol{X}^m$, it will increase the quantization error of $\boldsymbol{X} \setminus \boldsymbol{X}^m$, resulting in an accuracy drop across the LLaMA2-7B, LLaMA2-13B, and LLaMA3-8B. It is also worth

mentioning that the trend observed in the Mistral-7B-v0.3 model significantly differs from that of the LLaMA models. We believe this is primarily because, compared to the LLaMA models, the RH has effective in reducing the quantization error on $X^m$ as shown in Figure 13. Therefore, optimizing the quantization error of $X^m$ does not have a noticeable impact on the Mistral-7B-v0.3.

**Initialize with Randomized Orthogonal.** We conducted an ablation study on the use of RO with varying $\gamma$ values. From Figure 7 and Figure 8, it can be observed that, compared to using RH for initialization, our method achieved significant improvements in RO scenarios. However, due to the exceptional performance of RH, initialization and optimization using RH often yield superior final results compared to those obtained with random initialization.

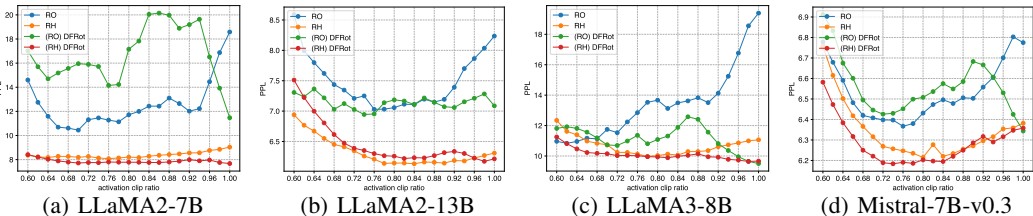

(a) LLaMA2-7B  (b) LLaMA2-13B  (c) LLaMA3-8B  (d) Mistral-7B-v0.3

Figure 9: Comparison of WikiText-2 perplexity results under different activation clip ratio for W4A4KV4. Weight is quantized via RTN.

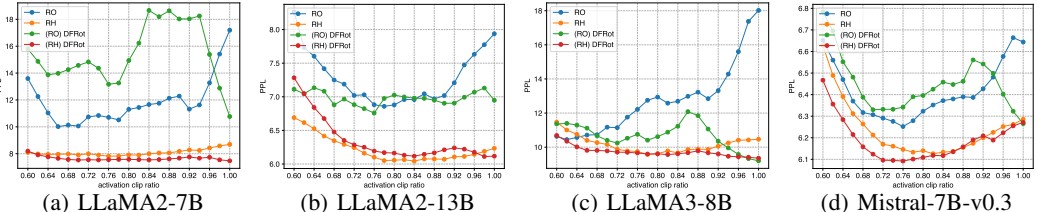

(a) LLaMA2-7B  (b) LLaMA2-13B  (c) LLaMA3-8B  (d) Mistral-7B-v0.3

Figure 10: Comparison of WikiText-2 perplexity results under different activation clip ratio for W4A4KV16. Weight is quantized via RTN.

**Ablation studies for activation clip ratio for RTN.** Activation clipping is a widely used quantization optimization technique, particularly effective for RTN. As shown in Figures 9 and 10, we conducted an experiment to investigate the effectiveness of DFRot for RTN quantization. The experimental results show DFRot always achieves better PPL at appropriate activation clip ratios. When the rotation matrix is initialized with RH, DFRot also achieves better results compared to RO. Additionally, we find that compared to GPTQ, which updates weights through compensation mechanisms, DFRot has a more pronounced effect on RTN quantization as it directly optimizes quantization errors. We believe that DFRot can further enhance the performance of methods like QServe, which do not incorporate GPTQ.

## 5 CONCLUSION

Eliminating outliers in LLMs through rotational invariance can significantly improve model quantization accuracy. In this paper, we find that in the context of 4-bit activation quantization, the fundamental reason for the difference in effectiveness between RO and RH is their performance on tokens with massive activations. Specifically, randomized Hadamard transformations perform better on these tokens. Based on this observation, we treat the problem as a long-tail optimization and construct a simple yet effective weighted quantization loss function to balance the importance of tokens. Furthermore, by alternately optimizing quantization parameters and employing orthogonal Procrustes transformations to refine the rotation matrix, our method, named DFRot, enhances the Rotated LLMs by achieving Dual Free, including *Outlier-Free* and *Massive Activation-Free*. DFRot significantly improves model accuracy in 4-bit activation quantization with just a single data sample and extra 8 minutes, achieving PPL improvements of 0.25 and 0.21 on W4A4KV4 and W4A4KV16, respectively, for the LLaMA3-8B, which is notable for its quantization challenges.

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

# A  QUANTIZATION ERROR FOR TOKENS WITH MASSIVE ACTIVATION IN LLAMA2-7B, LLAMA2-13B AND MISTRAL-7B-V0.3

More quantization results for LLaMA2-7B, LLaMA2-13B and Mistral-7B-v0.3:

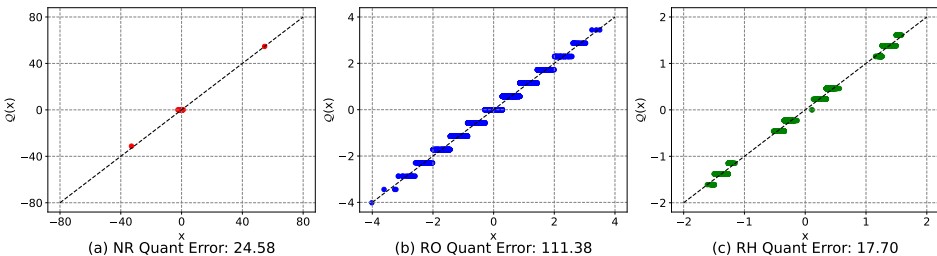

Figure 11: Comparison of 2D 4-bit quantization errors for tokens with NR, RO and RH for LLaMA2-7B from Figure 2.

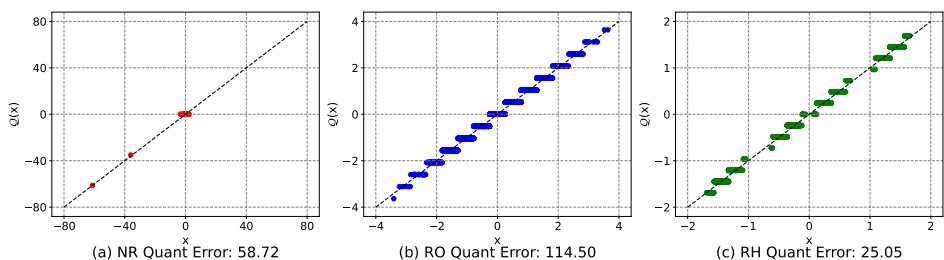

Figure 12: Comparison of 2D 4-bit quantization errors for tokens with NR, RO and RH for Mistral-7B-v0.3 from Figure 2.

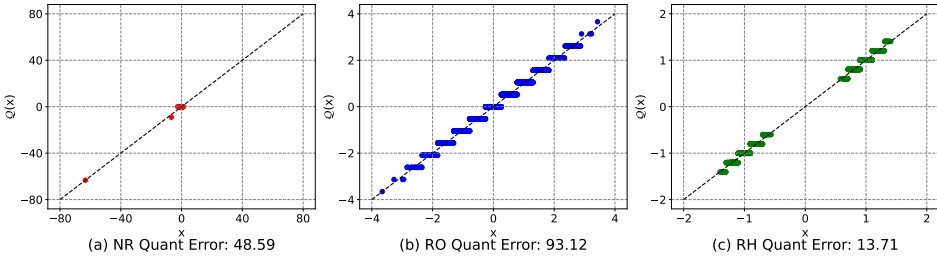

Figure 13: Comparison of 2D 4-bit quantization errors for tokens with NR, RO and RH for Mistral-7B-v0.3 from Figure 2.

# B QUANTIZATION ERROR BETWEEN VANILLA, RANDOM AND HADAMARD

More 2D quantization error visualization are shown as follows:

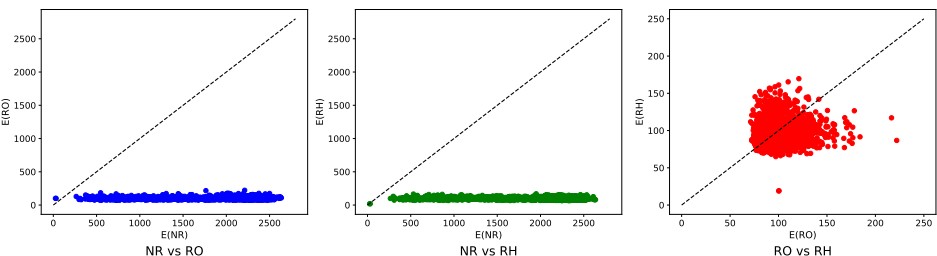

Figure 14: Comparison of 4-bit quantization error for the token with massive activation with NR, RO and RH for LLaMA2-7B from Figure 2.

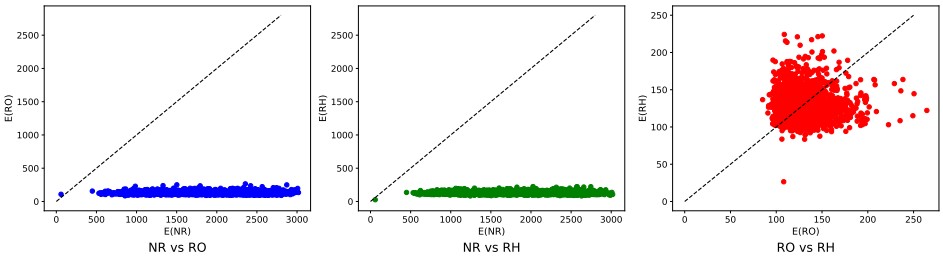

Figure 15: Comparison of 4-bit quantization error for the token with massive activation with NR, RO and RH for LLaMA2-13B from Figure 2.

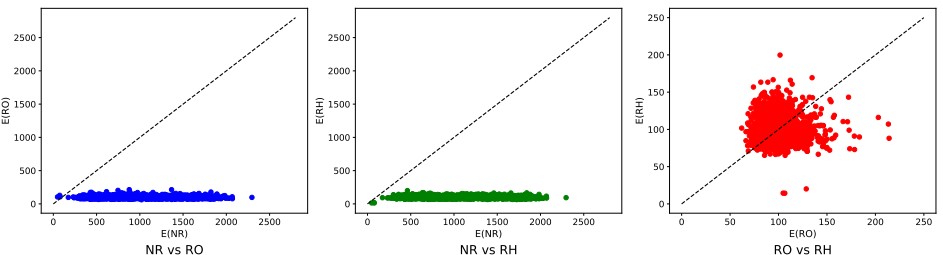

Figure 16: Comparison of 4-bit quantization error for the token with massive activation with NR, RO and RH for Mistral-7B-v0.3 from Figure 2.

Table 4: WikiText-2 perplexity (↓) results for LLaMA2-7B. The 4-4-4 and 4-4-16 represent W4A4KV4, W4A4KV16, respectively. We show the failed GPTQ experiments using NaN and the perplexity results>100 by Inf.

| Method | LLaMA2-7B | | Method | LLaMA2-7B | |
|---|---|---|---|---|---|
| Baseline | 5.47 | | Baseline | 5.47 | |
| | 4-4-4 | 4-4-16 | | 4-4-4 | 4-4-16 |
| RTN | NaN | NaN | GPTQ | NaN | NaN |
| QuaRot-RTN | 9.04 | 8.69 | QuaRot-GPTQ | 6.27 | 6.20 |
| SpinQuant-RTN | 6.20 | 6.17 | SpinQuant-GPTQ | 5.94 | 5.91 |
| OFMAF-RTN | 7.68 | 7.47 | OFMAF-GPTQ | 6.21 | 6.14 |

Table 5: Zero-shot accuracy (↑) of LLaMA2-7B with GPTQ on PIQA (PQ), WinoGrande (WG), HellaSwag (HS), Arc-Easy (A-e), Arc-Challenge (A-c), and LAMBADA (LA).

| Model | Method | W-A-KV | PQ | WG | HS | A-e | A-c | LA | Avg. |
|---|---|---|---|---|---|---|---|---|---|
| | FP16 | 16-16-16 | 79.11 | 68.98 | 75.99 | 74.54 | 46.42 | 73.88 | 69.82 |
| | QuaRot | 4-4-16 | 76.06 | 65.67 | 73.00 | 69.82 | 42.24 | 69.42 | 66.03 |
| | | 4-4-4 | 76.33 | 64.96 | 72.69 | 68.60 | 41.64 | 68.58 | 65.47 |
| LLaMA2-7B | SpinQuant | 4-4-16 | 75.24 | 66.14 | 72.82 | 68.77 | 40.44 | 70.88 | 65.72 |
| | | 4-4-4 | 76.66 | 65.98 | 72.78 | 70.92 | 42.06 | 70.12 | 66.42 |
| | DFRot | 4-4-16 | 77.15 | 65.82 | 73.17 | 69.78 | 44.37 | 70.66 | 66.83 |
| | | 4-4-4 | 76.22 | 64.96 | 72.41 | 70.75 | 42.66 | 69.92 | 66.15 |

## C    COMPARE TO SPINQUANT

Here, we present a detailed comparison between DFRot and SpinQuant (Liu et al., 2024):

- **Motivation.** The motivations behind SpinQuant and DFRot are entirely different. SpinQuant maintains the orthogonality of matrices throughout the training process using Cayley Optimization (Li et al., 2020), representing an end-to-end approach. In contrast, DFRot finds the fundamental reasons for performance differences in RO and RH is the quantization errors of tokens with massive activation. Recognizing the rarity of such tokens, it considers this a long-tail optimization problem and introduces a weighted loss function.

- **Optimization Methods.** SpinQuant optimizes rotation matrices using Cayley optimization, which necessitates loading the entire model and completing both forward and backward to obtain gradients during the training process. In contrast, DFRot regards the optimization of rotation matrices and quantization parameters as an implementation of an Expectation-Maximization (EM) like algorithm, employing Procrustes transformation to solve it, requiring only a single forward.

- **Optimization Cost.** To load and train the LLM, an NVIDIA A100 GPU with 80GB is almost essential for SpinQuant. In contrast, DFRot has lower hardware requirements than SpinQuant and can even optimize on RTX4090 24GB. For the training time, as metioned by SpinQuant, it takes ∼1.39 hours for LLaMA-3 8B, ∼1.25 hours for the LLaMA-2 7B, ∼2.36 hours for LLaMA-2 13B on 8 NVIDIA A100 GPUs. However, our DFRot only take ∼8 minutes for the LLaMA2-7B, ∼20 minutes for the LLaMA-2 7B, ∼8 minutes for LLaMA-2 13B on 1 NVIDIA A100 GPU. Therefore, DFRot is more efficient.

- **Performance.** Benefit from fine-tuning rotation matrices across the entire network through gradients, SpinQuant outperforms DFRot on the WikiText-2 PPL, as shown in Table 4, particulary in RTN quantization. However, we find for zero-shot tasks, DFRot still performs on par with SpinQuant as seen in Table 5. This indicates that the model's zero-shot capability does not have a direct correlation with its performance on the calibration dataset. By implementing *Outlier-Free* and *Massive Activation-Free*, DFRot also effectively enhances the performance of quantized LLMs. On the other hand, the goal of DFRot is not to achieve state-of-the-art performance. In contrast, it aims to highlight the significant importance of tokens with massive activation and explains the fundamental reasons why RH performance better than RO. Based on this finding, we propose an efficient and feasible solution to address the problem.

## D CALIBRATION DATA

In this section, we explain the reason why we only used a single data sample to calibrate the rotation matrix $\boldsymbol{R}_1$ in DFRot, and don not attempt to use more data:

- In LLMs, outliers and massive activations often appear in some fixed channels. Therefore, the process of optimizing the rotation matrix can be seen as an optimization of the distribution patterns of outliers and massive activations. We have simply use ten samples to calibrate the rotation matrix for LLaMA2-7B, but no significant improvement in accuracy was observed.

- Our calibration data is a sample with a length of 2048 tokens. Since we obtain the calibration set from each MHA and FFN, taking LLaMA2-7B as an example, we can obtain $2048 \times 32 \times 2 = 131072$ tokens as calibration tokens. This is relatively sufficient to statistically analyze the distribution patterns of outliers and massive activations.

## E ALGORITHM

---

**Algorithm 1** Optimization of Quantization Parameters and Rotation Matrix

---

**Require:** Token $\boldsymbol{x}$, initial rotation matrix $\boldsymbol{R}_1$, quantization function $\mathcal{Q}$
**Ensure:** Optimized rotation matrix $\boldsymbol{R}_1$ and quantization parameters $\boldsymbol{\eta_x}$
 1: Initialize $\boldsymbol{R}_1$ with randomized Hadamard matrix, $t = 0$
 2: **while** not converged **do**
 3:   // Step 1: Optimize Quantization Parameters $\boldsymbol{\eta_x}$
 4:   **for** each token $\boldsymbol{x}$ **do**
 5:     Compute quantization parameters $s, z$ via $\arg\min_{s,z} \|\boldsymbol{x}\boldsymbol{R}_1^{t-1} - \mathcal{Q}(\boldsymbol{x}\boldsymbol{R}_1^{t-1}, s, z)\|_2^2$
 6:     Update $\boldsymbol{\eta_x}^t = \mathcal{Q}(\boldsymbol{x}\boldsymbol{R}_1^{t-1}, s^t, z^t)$
 7:   **end for**
 8:   // Step 2: Optimize Rotation Matrix $\boldsymbol{R}_1$
 9:   Solve the Procrustes problem to update $\boldsymbol{R}_1^t$: $\boldsymbol{R}_1^t = \arg\min_{\boldsymbol{R}} \|\boldsymbol{X}\boldsymbol{R} - \boldsymbol{\eta_X}^t\|_F^2$
10:   $t = t + 1$
11: **end while**
12: **return** Optimized $\boldsymbol{R}_1^*$

---

## F RESULTS FOR QWEN2-7B

To further investigate the significance of massive activation on the final performance of the model, we conducted experiments using the recently renowned open-source model QWen2-7B. We find that the QWen2-7B model exhibits several different properties compared to LLaMA2-7B, LLaMA2-13B, LLaMA3-8B, and Mistral-7B-v0.3:

**Language Generation Task and Zero-Shot tasks.** Compared Table 1 to Table 6, when we used QuaRot.FP16() to retain the tokens with massive activation in FP16, although both of the performance of the RO and RH improved, the performance of RH still surpassed that of RO, which is inconsistent with the results in Table 1. For language generation task, similar to Mistral-7B-v0.3, DFRot does not achieve PPL improvement for QWen2-7B as shown in Table 7. However, from Table 8, we find it still improves accuracy for zero-shot tasks, which demonstrates the effectiveness of DFRot again.

**Quantization error and performance improvement.** We visualize the quantization error for QWen2-7B. As shown in Figure 17 and Figure 19, compared to previous models, QWen2-7B exhibits massive activation across multiple dimensions, which leads to a larger quantization error for the previous model. Based on this, both RO and RH effectively reduce the quantization error for tokens with massive activation, *e.g.* there is no red point in Figure 17 for QWen2-7B. This also explains why the PPL improvement of RO after using QuaRot.FP16() is not as pronounced as in previous models. Additionally, by comparing the quantization error between RO and RH in Figure 18, we observe that for QWen2-7B, the quantization error of RH slightly outperforms that of RO. Therefore, the performance of (RH) QuaRot.FP16() still surpasses that of (RO) QuaRot.FP16() .

**Quantize KV-Cache to 4-bit.** We find QWen2-7B is highly sensitive to the quantization of KV-Cache. When KV-Cache is quantized to 4 bits, the model performance completely collapses, even with W4A8KV4, which is significantly different from previous models. We find that this is due to QWen2-7B employs bias for Q, K, V module and some biases is large. This can lead to significant outliers for some specific channels and result in severe quantization errors for the KV-Cache quantization, even with rotation. Exploring how to better integrate rotation matrices with smooth methods for the quantization of KV-Cache is also an important research direction.

Table 6: WikiText-2 perplexity (↓) results for RO and RH for QWen2-7B. The 4-4-4, 4-4-16, 4-8-16 represent W4A4KV4, W4A4KV16, W4A8KV16 respectively. We show the perplexity results>100 by Inf. QuaRot.FP16() denotes retaining tokens with massive activations as FP16.

| Method | QWen2-7B | | | |
|---|---|---|---|---|
| | 4-4-4 | 4-4-8 | 4-4-16 | 4-8-16 |
| GPTQ | Inf | Inf | Inf | 7.57 |
| (RO) QuaRot | Inf | 8.07 | 8.07 | 7.25 |
| (RO) QuaRot.FP16() | Inf | 7.98 | 7.97 | - |
| (RH) QuaRot | Inf | 7.95 | 7.95 | 7.24 |
| (RH) QuaRot.FP16() | Inf | 7.91 | 7.91 | - |

Table 7: WikiText-2 perplexity (↓) results for QWen2-7B. The 4-4-4, 4-4-8, 4-4-16 represent W4A4KV4, W4A4KV8, W4A4KV16 respectively. We show the perplexity results>100 by Inf.

| Method | QWen2-7B | | | Method | QWen2-7B | | |
|---|---|---|---|---|---|---|---|
| Baseline | 7.14 | | | Baseline | 7.14 | | |
| Extra Time | +6min | | | Extra Time | +6min | | |
| | 4-4-4 | 4-4-8 | 4-4-16 | | 4-4-4 | 4-4-8 | 4-4-16 |
| RTN | Inf | Inf | Inf | GPTQ | Inf | Inf | Inf |
| QuaRot-RTN | Inf | 8.41 | 8.41 | QuaRot-GPTQ | Inf | 7.95 | 7.95 |
| DFRot-RTN | Inf | 8.40 | 8.43 | DFRot-GPTQ | Inf | 7.96 | 7.94 |

Table 8: Zero-shot accuracy (↑) of QWen2-7B with GPTQ on PIQA (PQ), WinoGrande (WG), HellaSwag (HS), Arc-Easy (A-e), Arc-Challenge (A-c), and LAMBADA (LA).

| Model | Method | W-A-KV | PQ | WG | HS | A-e | A-c | LA | Avg. |
|---|---|---|---|---|---|---|---|---|---|
| | FP16 | 16-16-16 | 81.07 | 72.45 | 78.83 | 74.66 | 49.83 | 71.82 | 71.44 |
| QWen2-7B | QuaRot | 4-4-16 | 78.02 | 68.11 | 75.16 | 72.22 | 45.56 | 66.83 | 67.65 |
| | | 4-4-8 | 78.02 | 66.38 | 75.24 | 71.34 | 46.76 | 67.13 | 67.48 |
| | | 4-4-4 | 57.18 | 49.09 | 28.56 | 31.99 | 25.94 | 0.45 | 32.20 |
| | DFRot | 4-4-16 | 78.73 | 69.30 | 75.59 | 74.12 | 49.40 | 67.63 | 69.13 |
| | | 4-4-8 | 78.51 | 66.93 | 75.06 | 72.18 | 49.06 | 66.85 | 68.10 |
| | | 4-4-4 | 55.88 | 49.17 | 27.79 | 34.34 | 25.60 | 0.50 | 32.21 |

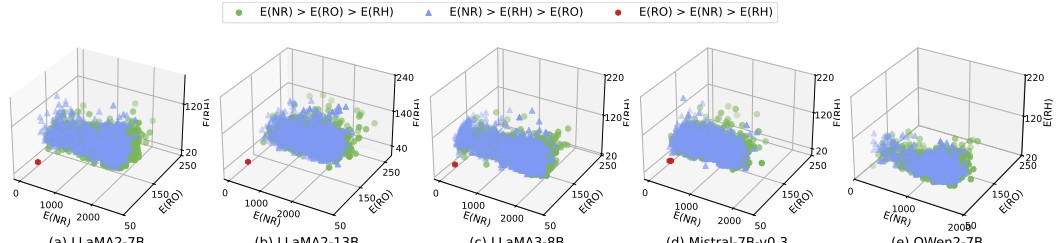

Figure 17: Comparison of 4-bit activation quantization error $E(\cdot)$ for each token with NR, RO and RH for (a) LLaMA2-7B, (b) LLaMA-2-13B, (c) LLaMA3-8B and (d) Mistral-7B-v0.3, (e) QWen2-7B. The tokens are from model.layers.6.post_attention_layernorm. Best viewed in color.

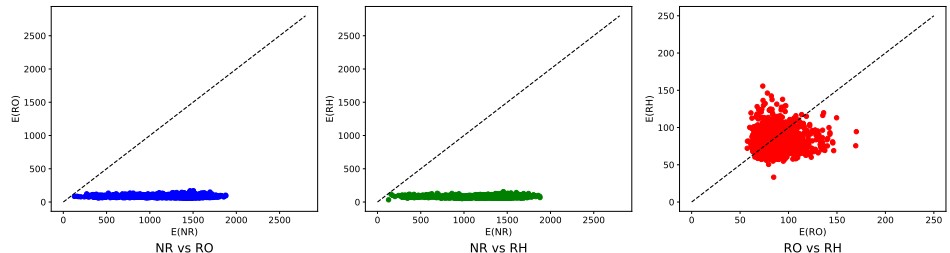

Figure 18: Comparison of 2D 4-bit quantization errors for tokens with NR, RO and RH for QWen2-7B.

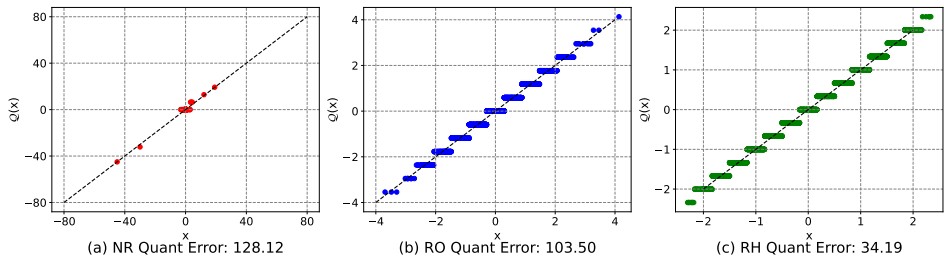

Figure 19: Comparison of 4-bit quantization error for token with massive activation without rotation (Vanilla), with RO and RH for QWen2-7B.

# G COMPARE WITH DUQUANT

Difference $R_1$ between QuaRot and DuQuant:

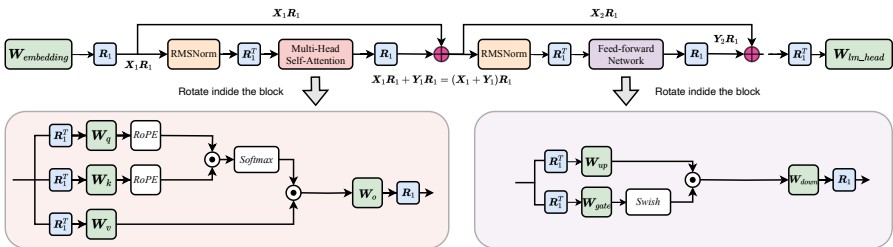

Figure 20: Computational graph for QuaRot.

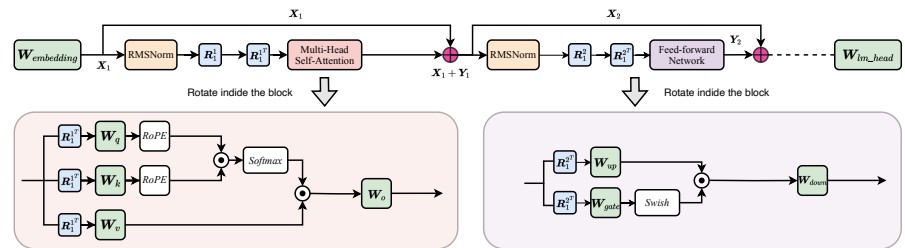

Figure 21: Computational graph for DuQuant.

# H VISUALIZATION FOR DIFFERENT LAYERS

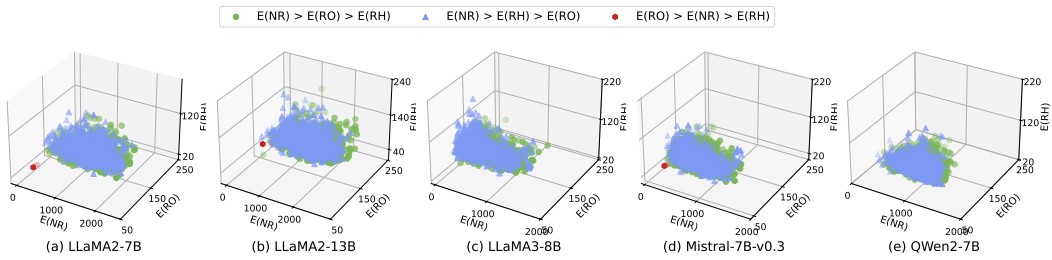

Figure 22: The tokens are from model.layers.2.input_layernorm

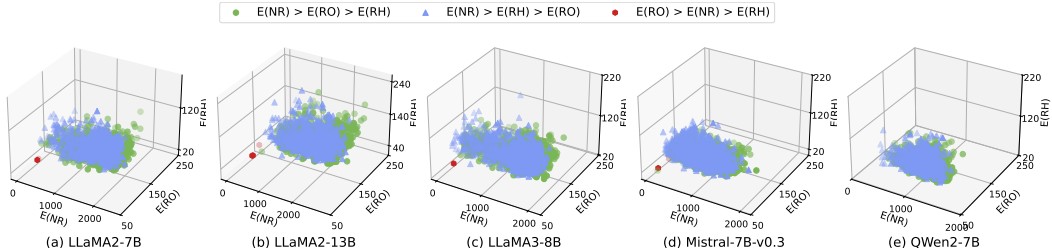

Figure 23: The tokens are from model.layers.2.post_attention_layernorm

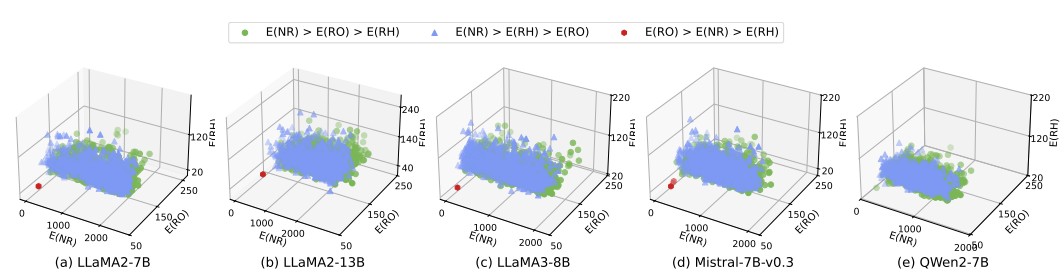

Figure 24: The tokens are from model.layers.5.input_layernorm

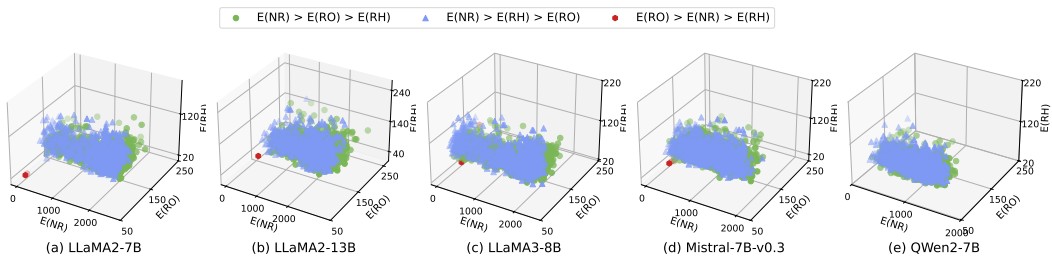

Figure 25: The tokens are from model.layers.5.post_attention_layernorm

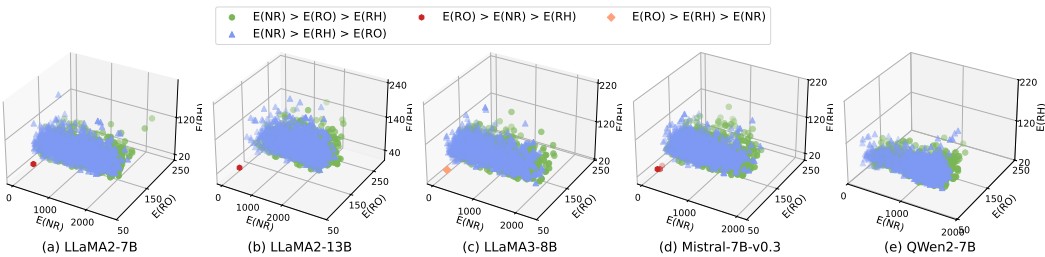

Figure 26: The tokens are from model.layers.7.input_layernorm

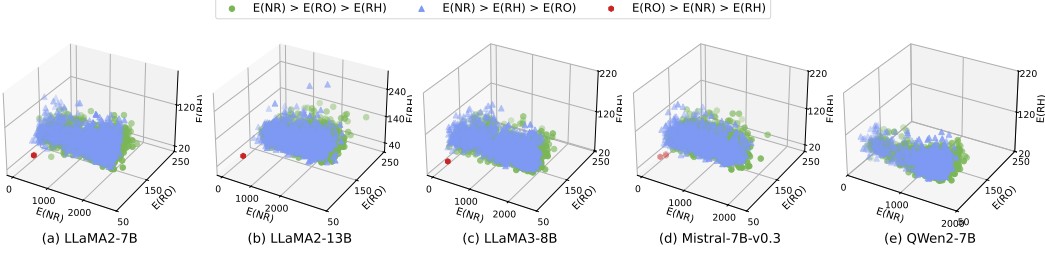

Figure 27: The tokens are from model.layers.7.post_attention_layernorm

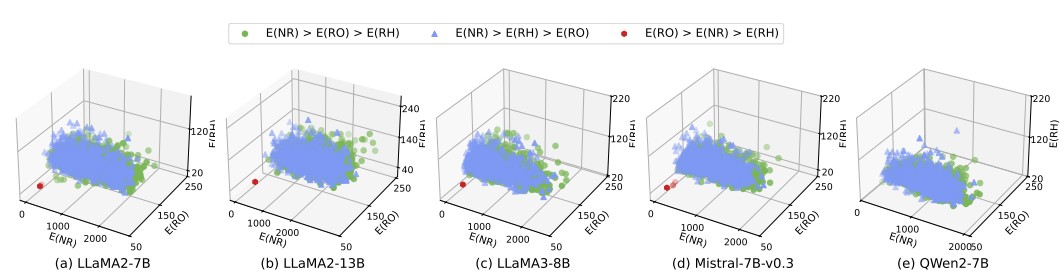

Figure 28: The tokens are from model.layers.9.input_layernorm

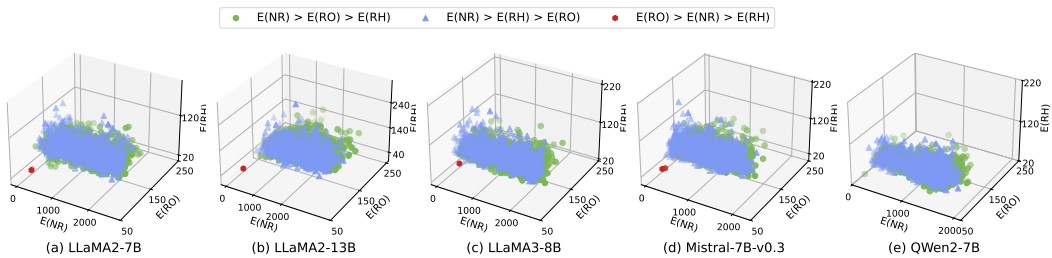

Figure 29: The tokens are from model.layers.9.post_attention_layernorm

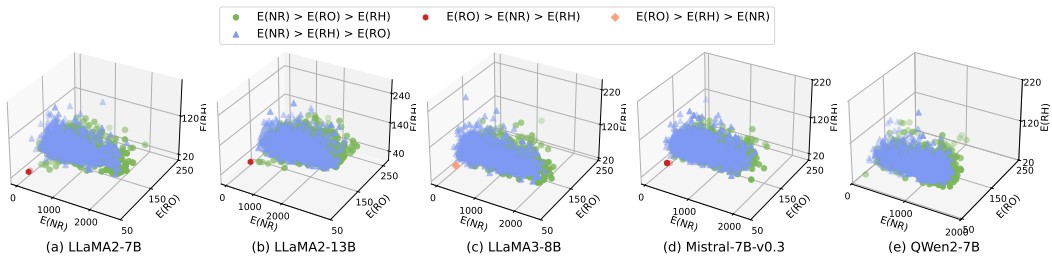

Figure 30: The tokens are from model.layers.11.input_layernorm

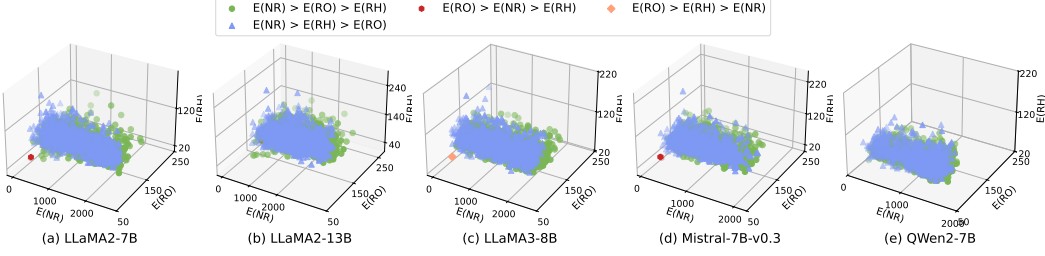

Figure 31: The tokens are from model.layers.11.post_attention_layernorm

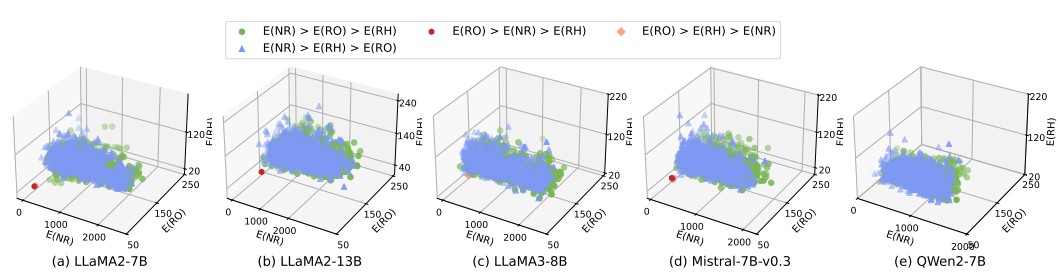

Figure 32: The tokens are from model.layers.13.input_layernorm

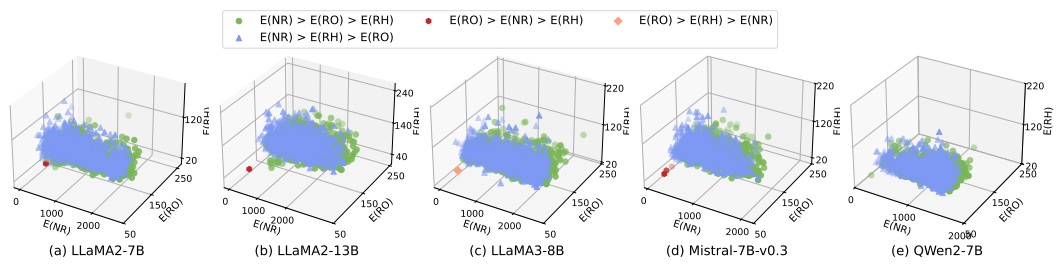

Figure 33: The tokens are from model.layers.13.post_attention_layernorm

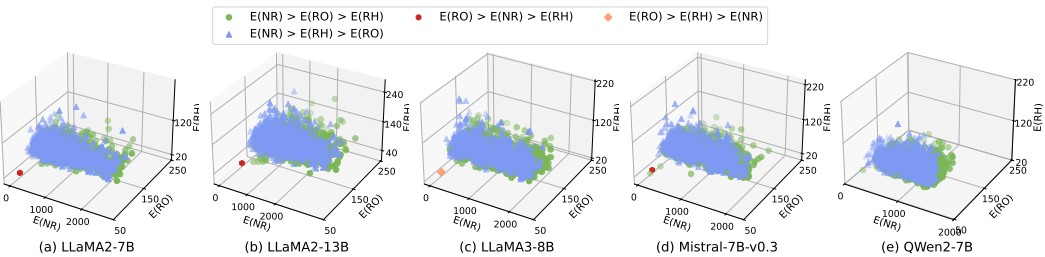

Figure 34: The tokens are from model.layers.15.input_layernorm

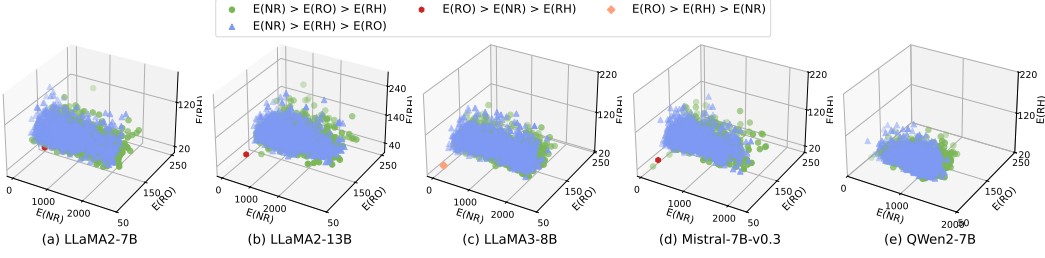

Figure 35: The tokens are from model.layers.15.post_attention_layernorm

# I  QUANTIZATION ERROR VISUALIZATION FOR DFROT

We show the quantization for LLaMA2-7B, LLaMA3-8B and Mistral-7B-v0.3 in Figure 36, Figure 37 and Figure 38 respectively. As seen, DFRot further reduces the quantization error of the token based on RH.

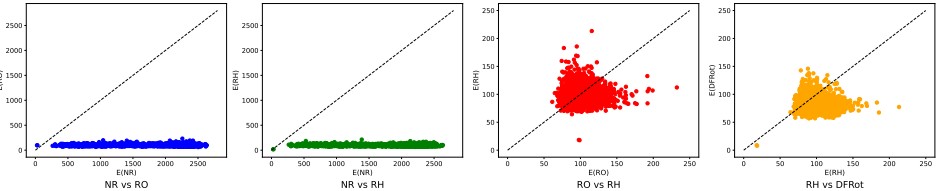

Figure 36: Comparison of 2D 4-bit quantization errors for tokens with NR, RO, RH and DFRot for LLaMA2-7B.

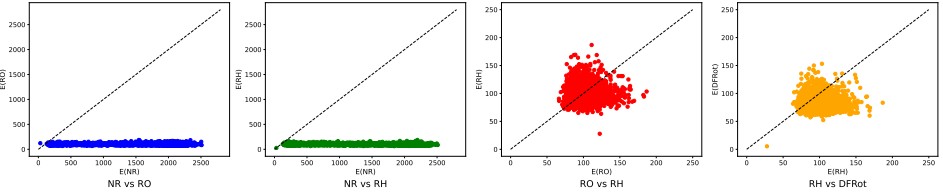

Figure 37: Comparison of 2D 4-bit quantization errors for tokens with NR, RO, RH and DFRot for LLaMA3-8B.

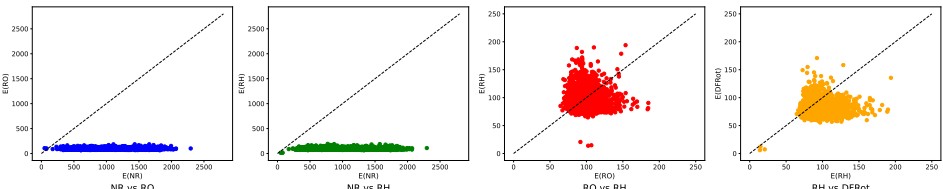

Figure 38: Comparison of 2D 4-bit quantization errors for tokens with NR, RO, RH and DFRot for Mistral-7B-v0.3.

