$\boldsymbol{W}$, resulting in a new weight matrix $\boldsymbol{R}_1^T \boldsymbol{W}$, which can be calculated offline. Since $\boldsymbol{R}_1^T \boldsymbol{R}_1 = \boldsymbol{I}$, the output from the linear layer remains unaffected. This *computational invariance* property of LLMs ensure the introduction of the rotation matrices without changing the original results.

A similar approach can be applied to rest layers within an LLM block. As shown in Figure 1, by transforming the weight matrices in the Multi-Head Attention (MHA) as $\boldsymbol{R}_1^T \boldsymbol{W}_q$, $\boldsymbol{R}_1^T \boldsymbol{W}_k$, $\boldsymbol{R}_1^T \boldsymbol{W}_v$, and $\boldsymbol{W}_o \boldsymbol{R}_1$, and the weights in the Feed-Forward Network (FFN) as $\boldsymbol{R}_1^T \boldsymbol{W}_{up}$, $\boldsymbol{R}_1^T \boldsymbol{W}_{gate}$, and $\boldsymbol{W}_{down} \boldsymbol{R}_1$, the hidden features within both MHA and FFN remain unchanged. Consequently, the output feature $\boldsymbol{Y}_1$ is transformed into $\boldsymbol{Y}_1 \boldsymbol{R}_1$, which will sum with the residual input $\boldsymbol{X}_1 \boldsymbol{R}_1$ satisfies $\boldsymbol{X}_1 \boldsymbol{R}_1 + \boldsymbol{Y}_1 \boldsymbol{R}_1 = (\boldsymbol{X}_1 + \boldsymbol{Y}_1)\boldsymbol{R}_1 = \boldsymbol{X}_2 \boldsymbol{R}_1$. The output will serve as the input for the next LLM block. Similarly, by transforming $\boldsymbol{W}_{lm\_head}$ to $\boldsymbol{R}_1^T \boldsymbol{W}_{lm\_head}$, the network output will remain unchanged.

Moreover, we can introduce additional rotation matrices to further mitigate outliers between layers. As illustrated in Figure 1, head-wise rotation matrices $\boldsymbol{R}_2$ and $\boldsymbol{R}_2^T$ can be applied to $\boldsymbol{W}_v$ and $\boldsymbol{W}_o$, while $\boldsymbol{

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

}_{\boldsymbol{x}} \left[ \left\| \boldsymbol{x}\boldsymbol{R}_1^{t-1} - \mathcal{Q}_{s,z}(\boldsymbol{x}\boldsymbol{R}_1^{t-1}) \right\|_2^2 \right], \boldsymbol{\eta}_{\boldsymbol{x}}^t \leftarrow \mathcal{Q}_{s^t,z^t}(\boldsymbol{x}\boldsymbol{R}_1^{t-1}) \tag{6}$$

In the case of dynamic asymmetric per-token quantization, we can independently determine the optimal quantization scheme for solving $s_{\boldsymbol{x}}$ and $z_{\boldsymbol{x}}$ for each $\boldsymbol{x}\boldsymbol{R}_1$:

$$\boldsymbol{\eta_x} = \mathcal{Q}_{\boldsymbol{g}}(\boldsymbol{x}\boldsymbol{R}_1) = \text{clamp}\left( \left\lfloor \frac{\boldsymbol{x}\boldsymbol{R}_1}{s} \right\rceil + z, 0, 2^N - 1 \right),$$

$$\text{where } s_{\boldsymbol{x}} = \frac{\alpha \max(\boldsymbol{x}\boldsymbol{R}_1) - \beta \min(\boldsymbol{x}\boldsymbol{R}_1)}{2^N - 1}, z_{\boldsymbol{x}} = -\left\lfloor \frac{\beta \min(\boldsymbol{x}\boldsymbol{R}_1)}{s_{\boldsymbol{x}}} \right\rceil \tag{7}$$

where $\lfloor \cdot \rceil$ indicates round operation, $N$ is the bitwidth, and $\alpha$ and $\beta$ is the clip ratio for upper bound and lower bound of quantization, respectively.

**Solving for $\boldsymbol{R}_1$.** Eq 5 is well-known as Procrustes problem (Mulaik, 2009). which involves finding the optimal rotation matrix $\boldsymbol{R}_1$ that best aligns two sets of points, minimizing the Frobenius norm of their difference. The solution to this problem can be obtained through Singular Value Decomposition (SVD). Specifically, given input matrices $\boldsymbol{X} = \{\boldsymbol{x}\}$ and its quantized version $\mathcal{Q}_{\boldsymbol{g}}(\boldsymbol{X})$ $= \{\mathcal{Q}_{\boldsymbol{g}}(\boldsymbol{x})\}$, the optimal $\boldsymbol{R}_1$ can be found:

$$\boldsymbol{R}_1 = \boldsymbol{U}\boldsymbol{V}^T, \text{where } \boldsymbol{U}, \boldsymbol{\Sigma}, \boldsymbol{V}^T = \text{SVD}(\boldsymbol{X}^T \mathcal{Q}_{\boldsymbol{g}^t}(\boldsymbol{X})). \tag{8}$$

where we treat the quantization parameters $\boldsymbol{g}^t$ as a constant.

**One-step optimization.** To find an improved rotation matrix $\boldsymbol{R}_1$ and quantization parameters $\boldsymbol{g}$, we perform the iterative process shown in Eq 4 and Eq 5 with just one round, which already yields significantly better performance, as demonstrated in the evaluation (Section 4). Specifically, a calibration set $\boldsymbol{X}^{cal}$ is randomly sampled from $\boldsymbol{X}$, the iterative process can be specified as:

$$s^t, z^t \leftarrow \arg\min_{s,z} \sum_{\boldsymbol{x} \in \boldsymbol{X}^{cal}} \left[ \left\| \boldsymbol{x}\boldsymbol{R}_1^{t-1} - \mathcal{Q}_{s,z}(\boldsymbol{x}\boldsymbol{R}_1^{t-1}) \right\|_2^2 \right], \boldsymbol{\eta}_{\boldsymbol{x}}^t \leftarrow \mathcal{Q}_{s^t,z^t}(\boldsymbol{x}\boldsymbol{R}_1^{t-1}), \tag{9}$$

then the resulting quantization parameters will be used to produce the rotation matrix:

$$\boldsymbol{R}_1^t \leftarrow \arg\min_{\boldsymbol{R}_1} \sum_{\boldsymbol{x} \in \boldsymbol{X}^{cal}} \left[ \left\| \boldsymbol{x}\boldsymbol{R}_1 - \boldsymbol{\eta}_{\boldsymbol{