# OpenReview forum: "DFRot: Achieving Outlier-Free and Massive Activation-Free for Rotated LLMs with Refined Rotation"
_ICLR.cc/2025/Conference — Submitted to ICLR 2025_

### Official Review · Reviewer_GJ5A · 2024-10-16

**Soundness:** 2
**Presentation:** 3
**Contribution:** 2
**Rating:** 3
**Confidence:** 5

**Summary:**

In this paper, the authors proposed a rotation-based post-training quantization (PTQ) scheme for the quantization of LLMs.
First, the authors empirically analyzed why random Hadamard transforms lead to better performance compared to random orthogonal transforms; the key reason is that random Hadamard transforms handle massive outliers much better than random orthogonal transforms.
Second, the authors presented how to optimize the random matrix that leads to the effective quantization of massive outliers; alternating optimization of quantization parameters and the rotation matrix has been presented.
Finally, the efficacy of the proposed method has been demonstrated via extensive experiments.

**Strengths:**

1. The authors analyzed why random Hadamard transforms lead to better performance compared to random orthogonal transforms.
2. The proposed weighted loss function is well-motivated and reasonable based on the above finding.
3. The claims presented by the authors are well supported by experimental results.

**Weaknesses:**

1. Some statements contradict each other (see Questions below).
2. Experiments are limited to some specific cases.
  - The size of LLMs is larger than 7B for which the performance gap between the conventional and proposed methods is not that noticeable.
  - The utilized weight quantizers (RTN and GPTQ) are not state-of-the-art.
  - The activation quantization configuration is dynamic per-token quantization, which is very difficult and expensive to implement in real hardware.

**Questions:**

1. Clarity on the proposed method and experimental setup
  - When measuring the quantization error in Section 3.2, the l2-norm of the activation perturbation (Eq.(1)) was used? If yes, the reviewer thinks that the final task loss degradation, rather than the activation quantization error itself, needs to be used as a metric.
  - How to determine the clip ratio $\alpha$ and $\beta$ has not been mentioned in Section 3.4.
  - When optimizing the rotation matrix (with fixed quantization parameters), it seems that the proposed weighted loss function in Eq.(2) has NOT been used. The corresponding solutions in Eqs. (8) and (10) consider all activations equally, regardless of whether activations being normal activations or massive outliers. In such a case, the authors did not consider massive outliers as in existing works.
  - While the authors mentioned that they performed the iterative optimization process with just one round (line 351), the number of iterations has been specified as 100 in line 371.

2.  More experiments are needed.
  - The size of LLMs used in the experiments is larger than 7B for which the performance gap between the conventional and proposed methods is not that noticeable for W4A4. The reviewer recommends conducting experiments with OPT models whose size varies from 125M to 66B.
  - From Table 2, the performance gap between QuaRot and DFRot reduces greatly when the weight quantizer changes from RTN to GPTQ. The reviewer thinks that if a better weight quantizer (e.g., aespa [1]) has been used, the gap between QuaRot and DFRot is very marginal. Please show the results with better weight quantizers.
  - From Figures 9 and 10, when the activation clipping is used, the performance gap between DFRot and no rotation is reduced. The reviewer thinks that the gap between QuaRot and DFRot becomes smaller if the activation clipping is used. Please show the results when applying activation clipping to both QuaRot and DFRot.
  - While most conventional works consider the per-token dynamic activation quantization, such a setting is expensive to implement in real-world scenarios. Please show the results for the **per-tensor static** activation quantization.

[1] Towards next-level post-training quantization of hyper-scale transformers, arXiv 2024.

---

> ### Author Response · Authors · 2024-11-18
> **Rebuttal for Reviewer GJ5A**
>
> **Q1**: The activation quantization configuration is dynamic per-token quantization, which is very difficult and expensive to implement in real hardware.
>
> **A1**: I would like to correct the author's misperception - in fact current inference frameworks (e.g. vllm) still use dynamic per-token quantization for $X$. In fact, compared to per-tensor static quantization, per-token's dynamic quantization with the help of operator fusion only produces a inference drop of about 5% in vllm, but the resulting accuracy gain is significant.
>
>
>
> **Q2**: Please show the results when applying activation clipping to both QuaRot and DFRot.
>
> **A2**: The reviewer seems to have misinterpreted the contents of Figures 9 and 10. Figures 9 and 10 show exactly the experimental comparisons about RO, RH and optimized RO and RH.
>
>
> **Q3**: Some concern about iteration round and the proposed weighted loss function in Eq.(2).
>
> **A3**: We performed 100 alternating optimizations of the alternating optimization process.
>
> For weight quantization error, we first separated and replicated $X$ in the actual calculation, a toy example is here: for $X=[x_1,x_2,x_3,x_4]$ and $x_4$ is the token with massive activation, if $\gamma=1$: $[x_1,x_2,x_3,x_4]$; $\gamma=2$: $[x_1,x_2, x_3, x_4, x_4]$.
>
> We will correct both of them. Many thanks to the reviewer for his/her careful reading and suggestions!
>
> **Q4**: How to determine the clip ratio $\alpha$ and $\beta$ has not been mentioned in Section 3.4.
>
> **A4**: We mention clip ratio $\alpha$ and $\beta$ in L366. In fact, the choice of clip ratio is more of a trick for quantization, and clip ratio is not significantly related to the main idea of this paper. Therefore, we set it to 1 to ensure the simplicity of the algorithm without any special explanation.
>
> **Q5**: The reviewer recommends conducting experiments with OPT models whose size varies from 125M to 66B.
>
> **A5**: The OPT model is now completely obsolete, and the current papers on new LLM quantification have almost ceased to explore the quantification of OPT. Therefore, I don't think additional experiments with the OPT model are necessary.
>
> We have added experiments with QWen2-7B to the Appendix. Although QWen2-7B does not have a red dot in the graph of quantization error (i.e., RO does not increase the quantization error of tokens with Massive Activation), we can still see from Figure 19 that QWen2-7B also has Massive Activation, and the data in Table 6 is also consistent with our conclusion in Section 3.2.
>
> We have explored all four LLMs, LLaMA2, LLaMA3, Mistral, and QWen2, in our paper, and our experiments are sufficient in terms of model diversity. If we have more computational resources in the future, we are more than happy to extend our experiments further.
>
>
> **Q6**: Author should show the results with better weight quantizers and the per-tensor static activation quantization.
>
> **A6**: From the ReviewerGuide, https://iclr.cc/Conferences/2025/ReviewerGuide, Review should know aespa to be our counterpart to contemporaneous work, so the use of aespa in the paper is not mandatory. At the same time aespa is not open source, and reproducing aespa is and is not mandatory for us.
>
> We would be happy to further explore the role of rotation in static quantization scenarios in the future, but for the purposes of this paper, static per -tensor quantization is not relevant to the main idea of the research in this paper. Referring to the ReviewerGuide,  https://iclr.cc/Conferences/2025/ReviewerGuide, we similarly argue that adding static quantization to the experiment is not necessary.

---

> > ### Comment · Reviewer_GJ5A · 2024-11-18
> > **Response to Rebuttal**
> >
> > Thanks to the authors for their efforts to address my concerns. However, my concerns have not been addressed at all.
> >
> > 1. Results with better weight quantizers
> >  - The first version of aespa has been published on 14 Feb 2024, so it is NOT contemporaneous work.
> >  - The performance improvement is very marginal (Table 2) when GPTQ has been applied. When better quantizers are used, the performance gap would be very small. Although not showing the result with the better weight quantizers, the authors should address this concern.
> >
> > 2. Dynamic per-token quantization
> >  - Is dynamic per-token quantization supported on NPUs (e.g., Qualcomm Hexagon)? If yes, please provide me the corresponding reference.
> >  - Why static per-tensor quantization is not relevant to the main idea of this paper? If massive activation outliers can be effectively suppressed by the proposed method, static per-tensor quantization performance would also be enhanced.
> >
> > 3. Experiments with models of various sizes
> >  - I admit that OPT models are out-dated. The main reason why I suggest the experiments with OPT is that all the models used for the validation have more than 7B parameters. I want to see the results for smaller models because for memory-limited devices (e.g., mobile), smaller models (e.g., 3B) have been employed usually.
> >
> > 4. Further comments
> >  - After reading comments from Reviewer 6qo1, I agree with the following reviewer's comment: it is trivial and unworthy of further investigation that RH can handle massive outliers better than RO.
> >  - While the authors provided some empirical results to show that RH indeed suppresses massive outliers better than RO, it is trivial phenomenon, considering the fact that the Hadamard matrix evenly distributes massive outliers across other channels while RO’s randomness does not guarantee the elimination of massive outliers.
> >
> > 5. Rating
> >  - When first reviewing this paper, my overall score for this paper was 4. However, the reviewing system did not allow me to provide 4, and that is why my initial score was 5. I still insist the overall score 4 due to the above reasons, and I hope Area Chairs to consider this when making a decision.

---

> > > ### Author Response · Authors · 2024-11-18
> > >
> > > Many thanks to the reviewers for their quick response, we would still like to summarize the conclusions of our work:
> > >
> > > Our work is not to show that RH suppresses massive outliers better than RO. However, **we conclude that this simple phenomenon is the root cause of the difference in performance between RH and RO.**
> > >
> > >
> > >
> > > That said, **quantifying that ~0.1% of the token can help us better understand the rotation**.
> > >
> > > As Reviewer MtcK says.
> > >
> > >     Overall I do like the story of this paper, which I see as trying to gain understanding between 2 rotation matrices, and using this understanding to improve the method.

---

> > > > ### Comment · Reviewer_GJ5A · 2024-12-02
> > > >
> > > > The authors fail to address my concerns about 1) integration with better weight quantizers, 2) per-tensor static quantization, and 3) experiments on various sizes of models, which the reviewer think necessary to demonstrate the efficiency of the proposed method.
> > > >
> > > > Thus, I cannot recommend the acceptance of this work and reduce the score to 3.

---

### Official Review · Reviewer_UiZw · 2024-11-02

**Soundness:** 3
**Presentation:** 2
**Contribution:** 2
**Rating:** 3
**Confidence:** 4

**Summary:**

The paper addresses the quantization problem for transformer architectures. It begins with an analysis of the impact of outliers—activations with unusually large values—on the performance of quantized models. The authors propose a commonly used approach of mitigate this issue: they rotate weight matrices and embedding vectors in the KV cache. The novelty of their approach is in using a learnable rotation matrix, designed to reduce quantization error specifically for outlier activations. This matrix is optimized via a cost function that prioritizes minimizing quantization error for these outliers. The authors introduce an iterative algorithm that appears computationally efficient for minimizing this cost function. The paper demonstrates the performance of the proposed method under 4-bit quantization in a series of experiments.

**Strengths:**

Quantization is a crucial problem, as memory constraints significantly limit the scalability of large language models. This paper tackles an important and timely research question, making it a compelling contribution to the field.

**Weaknesses:**

The presentation could be improved, as key details are missing that would help better position the work relative to existing methods and verify the authors' claims. For example, the paper does not adequately differentiate its method from SpinQuant and Quip, both of which also apply a learned rotation matrix to model weights before quantization.

Moreover, recent advancements in KV cache quantization are not sufficiently discussed. The practical performance of DFRot, for example, could have been compared to other recent approaches, such as:

- Liu, Zirui, et al. "Kivi: A tuning-free asymmetric 2bit quantization for kv cache." arXiv preprint arXiv:2402.02750 (2024).
- Zandieh, Amir, Majid Daliri, and Insu Han. "QJL: 1-Bit Quantized JL Transform for KV Cache Quantization with Zero Overhead." arXiv preprint arXiv:2406.03482 (2024).

**Questions:**

- What is the main difference between SpinQuant and Quip and your proposed method? Can you differentiate yourself from these methods?

- In Section 3.4, the authors mention that quantization is regarded as clustering, but the explanation is vague. Are the authors suggesting clustering the rows of matrix X? Each row is a high-dimensional vector, and clustering such vectors directly would likely yield random and meaningless results. Practical clustering-based quantization methods typically partition the coordinates of high-dimensional vectors into smaller blocks, applying clustering to these blocks. A more precise explanation here would improve clarity.

- The experiments are confusing. Which experiments focus on model quantization, and which are aimed at KV cache quantization?

- How does the proposed method perform in comparison to the aforementioned KV cache quantization methods?

Additionally, I would like to note: please try to address my concerns, and rest assured that I will consider raising my score if you provide sufficient results during the rebuttal.

---

> ### Author Response · Authors · 2024-11-14
> **Rebuttal for Reviewer UiZw-Part1**
>
> **Q1**: The paper does not adequately differentiate its method from SpinQuant and Quip.
>
> **A1**: Main difference between SpinQuant, Quip and DFRot:
>
> From a temporal perspective, QuIP was the first to be proposed. It improved the quantization of the LLM by enhancing the incoherence between the weights and the Hessian matrix by using the Kornecker matrix, but QuIP did not focus on the rotational invariance of the LLM
>
> QuaRot was the first to introduce the rotational invariance mentioned in the SliceGPT into the field of quantization of the LLM. Althugh it still used rotational matrix to eliminate outliers, the application of rotational invariance, which allows $R_1$ to be reparameterized to the network weights without introducing any additional computational overhead, is the core of QuaRot's real contributions.
>
> SpinQuant is a follow-up work based on QuaRot, and SpinQuant extends the orthogonal matrices in QuaRot to the training space, improving the effectiveness of the network. **We provide a comparison with SpinQuant in the Appendix.C and discuss in detail the differences between our work and SpinQuant.**
>
> The contribution and difference of our work (DFRot) consists of two parts:
>
> 1. Firstly, we rigorously explain the essential reason for the performance difference in 4-bit activation quantization between RO and RH--RH can solve the quantization of tokens with massive activation much better (this can be seen in Section 3.2), we get the conclusion that such tokens are both fewer and more important. This phenomenon has been overlooked by previous work, and we were the first to discover it, and we think it's a phenomenon of great significance.
> 2. Meanwhile, in Section 3.3, we further propose a matrix optimization method based on orthogonal Procrustes transform for the method in Section 3.2. Compared to SpinQuant, our method is more similar to PTQ, because SpinQuant needs to load the whole model for training (8 $\times$ A100 80G GPUs are used in SpinQuant's paper), whereas our scheme only needs to collect the calibration set for optimization, and our method is more efficient.
>
> **Q2**: About why we regarded quantization as clustering.
>
> **A2**: We use per-token dynamic quantization in our paper, as can be seen in Eq 7. In this case, we consider **the quantization result after per-token quantization to be the clustering center, e.g. each token has its own clustering center, not "clustering the rows of matrix $X$"**.
>
> The reason why we consider the optimization of Eq.1 as a clustering process motivated by the following considerations:
> 1. There are two optimization objectives in the problem, namely the quantization parameter and the rotation matrix $R_1$;
> 2. We solve the problem using an alternating approach.
>
> Therefore, under the condition of per-token dynamic quantization for $X$, we consider the quantization result after quantization of each token to be the clustering center, and the rotation matrix to be the mapping function of the clusters. Referring to the optimization of the clustering process: making the clustering center closer to the mapped data and optimizing the mapping function to make the data closer to the clustering center, we propose our optimization scheme.
>
>
> **Q3**: Some clarification about KVCache quantization.
>
> **A3**: We are very sorry that our presentation of the experiment made the reviewer confused. In fact, we do not study the quantization of KVCache in our work, and **the table showing W4A4KV4 and W4A4KV16 indicates that we applied the method to two different quantization configurations**. For the quantization of the 4-bit KVCache, we still follow QuaRot's scheme and use the Hadamard matrix to rotate PostRoPE features before quantization for a fair comparison with QuaRot.
>
> It is worth emphasizing that the study of our work is this part of the features of each Attention and FFN Block input, because after rotational invariance, this part of the features will change from an unrotated $X_n$ to $X_nR_1$. In other words, our research is independent of KVCache's compression, which focuses on different parts of features. Therefore, the two should be complementary.
>
> Since FlashAttentionV3 also uses Hadamard as the rotation matrix in the self-attention, we can know that understanding the role of rotation matrix in reducing the quantization error of KVCache, and further optimizing the rotation matrix for KVCache is also of great significance to the compression of KVCache, and in the future we will further related explorations.

---

> > ### Comment · Reviewer_UiZw · 2024-11-19
> > **Thanks for your response.**
> >
> > I am confused about what you mean by activation quantization? You method does not quantize the KV cache and it also does not quantize the model weights. You stated that you quantize the activations. Could you clarify what exactly does that mean?

---

> > > ### Author Response · Authors · 2024-11-20
> > > **Thanks for Reviewer UiZw response**
> > >
> > > I hope that my following explanation will solve your confusion:
> > >
> > > 1. **For activation quantization and kv cache quantization**: in LLM quantization, researchers distinguish between the study of quantization of activation values, i.e., the quantization of Activation is currently studied based on the Linear layer; whereas the quantization of KVCache is another direction of research, mainly in the field of long texts. In short, W4A4 is the abbreviation of W4A4KV16, if KVCache is to be quantized to 4bit, it should be W4A4KV4.
> > >
> > > 2. **For weght quantization**: In Section 3, what we are describing is that we found that this 0.1% token (activation) is the root cause that determines the difference in performance between RH and RO, so we optimized the rotation matrix based on this finding. For weight quantization, we simply used RTN and GPTQ, two methods well known in the field of model quantization, so we only declared their use in the experimental setup and did not describe them in the Section 3. It can be simply considered that DFRot builds on QuaRot by further understanding rotational invariance, the reasons for the success of rotation matrices, and identifying the flaws that exist and further optimizing the optimization matrices.

---

### Official Review · Reviewer_MtcK · 2024-11-03

**Soundness:** 3
**Presentation:** 2
**Contribution:** 3
**Rating:** 5
**Confidence:** 4

**Summary:**

The authors study the difference between two rotation matrices used for post training quantization, and shows some empirical evidence explaining their difference in performance. Specifically, the random hadamard transformation is better at reducing quantization error for tokens with massive activations in 4 bit activation quantization. Using these insights, the authors take the framework for inserting rotations into an LLM architecture proposed by QuaRot, and learn a subset of those rotation matrices. Their optimization uses a weighted sum between the tokens with massive activations and those without. Experiments are provided on Llama2 7b, 13b, llama3 8b, and mistral 7b.

**Strengths:**

- I think looking at the massive activations is a sensible thing for investigating RH vs RO.
- And I think that the resulting approach which takes advantage of these insights is interesting.
- Overall I do like the story of this paper, which I see as trying to gain understanding between 2 rotation matrices, and using this understanding to improve the method.

**Weaknesses:**

I have concerns about this paper. Overall, I think the writing could use work. It's also unclear why the authors only target the "R1" rotation matrices. But more significantly, I think they should include experiments comparing to SpinQuant. This is the more appropriate comparison, because this method also learns their rotation matrices. Finally, I think that there should be a inference time consequence to this method that I think the authors should make more clear.

Some other comments:
- I find the difference between random hadamard and random orthogonal confusing, because hadamard transform is orthogonal.
- I found the language describing the relative strengths/weaknesses of RO/RH (e.g. paragraph starting L77) to be confusing.
- I think the writing in Section 3 could be improved. I believe that 3.1 describes how to integrate rotation matrices into the architecture. But I think readers unfamiliar with QuaRot can be confused. I think that some summary sentences describing the overall goal, and pointing readers to QuaRot for further details, would be helpful. I see that QuaRot is cited, but only for a RMSNorm equality, it is not clear from the text that QuaRot actually describes the overall approach of a way to integrate rotations.
- I'm unconvinced by the line of argument in Section 3.2. The authors say that the activation quantization error in Fig 2 is not that different between "RH" and "RO", but the overall PPL from Table is significantly different. How Fig 2 is just from a single layer, and I honestly think it's hard to tell from the plots if there is a significant difference or not. Are there perhaps some summary statistics about activation errors that the authors can share?

**Questions:**

- One suggestion: hadamard transformation is orthogonal, and therefore I think the contrast of hadamard vs orthogonal is not entirely accurate and a bit confusing. Perhaps the first time this comparison is made specify this difference, and then I think it's ok as a shorthand to say hadamard vs orthogonal.
- Per Section 3.1, why do the authors only target rotation matrices R1? and not R2,R3,R4?
- Also, R1 will be applied to the input activations, and therefore will need to be applied during inference. What are the computational costs?
- What exactly are the RO matrices? RH have been proposed by QuIP#, but it is unclear what the RO matrices are.
- Another reason why I think the naming of "random orthogonal" matrices is misleading: the authors are making claims about RH vs RO performance, but as far as I can tell "RO" is simply the matrices generated by the SpinQuant paper. It's more accurate to say that the rotations matrices from these two papers are being compared.
- The authors provide an empirical observation, that RH is better at reducing the quantization error for tokens with massive activations in 4 bit activation quantization. But why is this the case?
- what does the "time" row in Tab 2 mean? is this training or inference?
- Why don't the authors compare to SpinQuant? Comparing Llama2-7b 4-4-4 Tab 1 in the SpinQuant paper to Tab2 in this paper, appears that SpinQuant achieves a better wikitext 2 ppl of 5.9, vs DFRot which achieves 6.2. If I've made an incorrect comparison, please let me know. The baselines between the two papers looked to be comparable.

---

> ### Author Response · Authors · 2024-11-14
> **Rebuttal for Reviewer MtcK Part-1**
>
> First of all, I am very grateful for the reviewer's comments, who really understand the motivation and contribution of our work!
>
> **Q1**:  It's also unclear why the authors only target the "$R_1$" rotation matrices.
>
> **A1**: This stems from some of the concerns we have in moving forward:
>
> 1. Based on the premise of rotational invariance proposed by QuaRot, the optimization of $R_1$ is the most challenging because $R_1$ runs through the whole network, while $R_2$, $R_3$, and $R_4$ are block wise, so we can finetune them by block by block, which is not difficult from the optimization point of view;
>
> 2. The most fundamental MOTIVATION of our work is based on the difference between RO and RH. In fact, the use of RO and RH in QuaRot is only in $R_1$, and does not extend to $R_2$, $R_3$, and $R_4$. Also, the study of tokens with Massive Activation mentioned in the paper is mainly in the position of $R_1$, and in fact the features appearing in the positions of $R_2$, $R_3$, and $R_4$ we found in our experiments that there are no such Massive Activation, this is because RMSNorm normalizes the features when they input in Attention and FFN block;
>
> 3. Within the context of ICLR's 10-page body, our optimization for $R_1$ (from explaining $R_1$'s advantages to proposing solutions) has reached the space limit. We are currently moving forward with optimizations for $R_2$, $R_3$, and $R_4$ to further understand the advantages of rotation, which is our future wage, and we very much hope that our work will provide some INSIGHT into the quantization of lower bits in LLM.
>
>
> **Q2**: Comparing to SpinQuant.
>
> **A2**: We provide a comparison with SpinQuant in the Appendix.C and discuss in detail the differences between our work and SpinQuant.
>
> **Q3**: There should be a inference time consequence to this method.
>
> **A3**: Many thanks to the reviewers for the suggestions. We did not include inference time considerations in the paper because:
> 1. We focused our work on explaining the reasons why RO and RH can produce performance differences, and then proposed a suitable scheme to optimize R1;
> 2. Our approach is a further deeper understanding and optimization of QuaRot, and does not modify the computational graph of the network based on QuaRot pairs. In other words, our inference speed is exactly the same as QuaRot, because the optimized R1 can also be fused into the existing weights of the network without introducing any extra overhead in inference.
>
> **Q4**: The difference between random hadamard and random orthogonal confusing, because hadamard transform is orthogonal.
>
> **A4**: I understand yourconcern. the $\text{RH}$ matrix is indeed equally orthogonal. We call it that because we refer to QuaRot's and SpinQuant's descriptions of these two matrices. Let me explain the process by which these two matrices are generated to understand the difference:
>
> $\text{RH}$: Given a Hadamard matrix H, a random binarization dignal matrix $A_{ii}$ in {$+1,-1$}, $M=HA$,so $({M})({M})^T=HAA^TH^T=I$,where $I$ is a IdentityMatrix.
>
> $\text{RO}$: $\text{RO}$ is obtained from QR decomposition, where $Q$ is orthogonal.
>
> We will do our best to illustrate the difference between these two matrices in a better way, although it is true that it is a bit of a slog, since from the point of view of the set, that $\text{RH} \in \text{RO}$.

---

> ### Author Response · Authors · 2024-11-14
> **Rebuttal for Reviewer MtcK-Part2**
>
> **Q5**: I found the language describing the relative strengths/weaknesses of RO/RH (e.g. paragraph starting L77) to be confusing.
>
> **A5**: I apologize for the confusion, indeed the description of L77 should be understood in the context of Figure 2 through Figure 4, and with some understanding of Massive Activation[1].
>
> Your further discussion is very welcome and will help us to further improve the quality of our work!
>
> **Q6**: Section 3 could be improved.
>
> **A6**: Since QuaRot applies a lot of space to describe rotational invariance, but due to the limited space in our paper, we simplify it while causing confusion for readers who are unfamiliar with QuaRot. Thank you very much for your suggestion and we will upgrade the description of Section 3.
>
> **Q7**: More explanation for say the activation quantization error in Fig 2 is not that different between $\text{RH}$ and $\text{RO}$.
>
> **A7**: In Figure 2 we visualize for different cases of quantization error, and we can intuitively see that the green and blue colors are close to 50% each, which also indicates that there is almost no significant difference between the quantization errors of RO and RH.
>
> In order to further demonstrate this, we show 2D visualization for Figure 2 in Figure 3 (LLaMA2-7B), Figure 11 to Figure 13 (LLaMA2-13B, LLaMA3-8B, Mistral-7B-v0.3).
>
> Take Figure 3 as an example, we compare quantization error for tokens between RO and RH in the 3rd figure (red one) and the dashed line is $y=x$. We can see from Figure 3 that the quantization errors of the vast majority of tokens are basically distributed on both sides of the dotted line without significant differences, which also corresponds to what we described in Section 3.2. Similar results can also be observed in Figure 11 to Figure 13.
>
> Meanwhile, we have applied Figure 2 to other layer pairs to visualize the quantization errors, as shown in Figure 20 to Figure 23. It can be obtained that the phenomenon similar to Figure 2 is reflected in other layers of the model, which also shows that our illustration is reasonable.
>
> **Q8**: $R_1$ will be applied to the input activations, and therefore will need to be applied during inference. What are the computational costs?
>
> **A8**: Not any additional. This is the core of rotational invariance and truly elegant. Thanks to the property $RMSNorm(XR_1) = RMSNorm(X)R_1$ (which can be obtained from QuaRot, SliceGPT), $R_1$ can be merged by reparameterization into weight, changing ($W_{embedding}$, $W_{lm\_head}$, $W_q$, $W_k$, $W_v$, $W_o$,
> $W_{up}$,$W_{gate}$, $W_{down}$) into ($W_{embedding}R_1$, $R_1^TW_{lm\_head}$, $R_1^TW_q$, $R_1^TW_k$, $R_1^TW_v$, $W_oR_1$,
> $R_1^TW_{up}$,$R_1^TW_{gate}$, $W_{down}R_1$). In this way, the block output such as $X_2=X_1R_1+Y_1R_1=(X_1+Y_1)R_1$. And network output will not change, i.e. introducing no additional cost during the inference. (L181-L188)
>
> **Q9**: The authors provide an empirical observation, that RH is better at reducing the quantization error for tokens with massive activations in 4 bit activation quantization. But why is this the case?
>
> **A9**: It is very difficult to analyze mathematically, but a toy example can be given here to illustrate the phenomenon:
>
> $$
> X=\left[\begin{array}{cccc}
>  0 ;
>  0  ;
>  0  ;
>  1
> \end{array}\right]
> $$
>
> $$
> R_1=\frac{1}{2}\left[\begin{array}{cccc}
> -1 & 1 & -1 & 1 ;
> -1 & -1 & -1 & -1 ;
> -1 & 1 & 1 & -1 ;
> -1 & -1 & 1 & 1
> \end{array}\right]
> $$
>
> $$
> R_2=\left[\begin{array}{cccc}
> -0.2379 & -0.9706 &  0.0170 &  0.0318 ;
>  0.8611 & -0.2279 & -0.2426 & -0.3843 ;
> -0.4434 &  0.0766 & -0.6055 & -0.6565 ;
>  0.0732 & -0.0099 & -0.7578 &  0.6483
> \end{array}\right]
> $$
>
> $$
> R_1X=\left[\begin{array}{cccc}
>  0.5 ;
> -0.5  ;
> -0.5  ;
>  0.5
> \end{array}\right]
> $$
>
> $$
> R_2X=\left[\begin{array}{cccc}
>  0.0318 ;
> -0.3843  ;
> -0.6565  ;
>  0.6483
> \end{array}\right]
> $$
>
> In this way, if we want to quantize activation to 1-bit, $X$ and $R_1X$ do not have quantization error and $R_2X$ has quantization error. In other words, the “random” pattern of random orthogonal matrices causes the token with massive activation to increase its quantization error after rotation.
>
> **Q10**: What does the "time" row in Tab 2 mean? is this training or inference?
>
> **A10**: Time refers to the time required for our method to optimize $R_1$, which can be interpreted as the extra time added to our method compared to RTN/GPTQ.
>
>
>
> [1] Sun, M., Chen, X., Kolter, J.Z. and Liu, Z., 2024. Massive activations in large language models. arXiv preprint arXiv:2402.17762.

---

> > ### Comment · Reviewer_MtcK · 2024-11-25
> > **Reviewer response**
> >
> > Thank you for the detailed response.

---

### Official Review · Reviewer_6qo1 · 2024-11-04

**Soundness:** 2
**Presentation:** 2
**Contribution:** 1
**Rating:** 3
**Confidence:** 4

**Summary:**

The paper proposes a method, DFRot, designed to address outliers encountered when quantizing large language models (LLMs). The method initializes with a Hadamard matrix and adapts to activation outliers. It is then applied to LLaMA/Mistral models under W4A4 settings to verify its effectiveness.

**Strengths:**

1. The paper is well-written and clearly structured, making it easy to follow.
2. The method’s motivation is substantiated by experimental results, lending it credibility.

**Weaknesses:**

1. The authors extensively emphasize that RH outperforms RO and effectively handles massive outliers, which RO cannot. However, this appears trivial and unworthy of further investigation. Firstly, it is evident that RH can distribute outliers evenly across channels, which RO cannot accomplish. Secondly, as already noted, numerous works, such as QuaRot [1], have employed Hadamard rotation matrices and confirmed the effectiveness of RH. Lastly, the paper lacks theoretical analysis on this point.
2. The novelty and effectiveness of the method seem limited. Numerous prior studies have explored learnable matrices, including SpinQuant [2] and DuQuant [3]. **There is a noticeable lack of comparison with these methods, both in terms of methodological differences and performance metrics.** From my perspective, the results in the paper do not outperform any of these works [2, 3]. Additionally, the model size used (≤13B) is relatively small, which limits the persuasiveness of the results.
3. The method does not include evaluations related to actual memory reduction or speedup.

[1] Ashkboos, S., Mohtashami, A., Croci, M. L., Li, B., Jaggi, M., Alistarh, D., ... & Hensman, J. (2024). Quarot: Outlier-free 4-bit inference in rotated llms. arXiv preprint arXiv:2404.00456.

[2] Liu, Z., Zhao, C., Fedorov, I., Soran, B., Choudhary, D., Krishnamoorthi, R., ... & Blankevoort, T. (2024). SpinQuant--LLM quantization with learned rotations. arXiv preprint arXiv:2405.16406.

[3] Lin, H., Xu, H., Wu, Y., Cui, J., Zhang, Y., Mou, L., ... & Wei, Y. (2024). Duquant: Distributing outliers via dual transformation makes stronger quantized llms. arXiv preprint arXiv:2406.01721.

**Questions:**

1. Is the rotation matrix a Diagnol Block matrix, or a full $D\times D$ matrix, where $D$ is the hidden dimension size?
2. How similar is the final matrix obtained by your method and the Hadamard matrix used for initialization? If the similarity is small, it suggests that the method’s effectiveness may rely more on the Hadamard matrix than on your training approach.
3.It would be helpful if the authors provided visualizations comparing 1) original activation; 2) activation transformed by QuaRot; 3) activation transformed by their method, to better demonstrate the effectiveness of trained matrix.
5. Do different layers or projections share a common rotation matrix? Will this method lead to increased memory or time costs during the inference phase?

---

> ### Author Response · Authors · 2024-11-14
> **Why DFRot didn't compare it to DuQuant, because they are completely different.**
>
> Although I am very grateful to the reviewers for reading our work, I have to admit that you have completely missed the difference between QuaRot and DuQuant. Although both methods appear on the surface to use a rotation matrix to eliminate outliers in the activation values of an LLM, this **QuaRot and DuQuant have huge difference**:
>
> 1. **Computational overhead**: QuaRot uses the same rotation matrix $R_1$ for the global network, which, thanks to the nature of the RMSNorm, allows $R_1$ to be fully reparameterized into the model's weights, and the introduction of $R_1$ doesn't introduce any additional computational overhead; DuQuant uses a different rotation matrix for each Block, which leads to the fact that after each RMSNorm we DuQuant uses a different rotation matrix for each Block, which leads to an online rotation of the features after each RMSNorm, which leads to a large number of floating-point matrix multiplications, and therefore I believe that DuQuant is not fully W4A4, and the authors do not explicitly discuss the problems that these floating-point computations pose to the design of a W4A4 inference system in the paper.
>
> 2. **Optimization Difficulty**: QuaRot uses a single $R_1$ for the entire network, so optimizing $R_1$ will be very difficult because we can't optimize $R_1$ using the Block by Block approach (i.e., the quantization result of the previous Block will change while optimizing the rotation matrix $R_1$ for the next Block optimization); but DuQuant uses a different $R_1$ for each Block, in a way that would make QuaRot's optimization difficulty completely non-existent, by simply optimizing it using only the Block by Block optimization method.
>
> **In summary, I don't think QuaRot and DuQuant can be compared**. This is because QuaRot and DuQuant are completely different from the point of view of inference reasoning and optimization, and DuQuant completely loses the advantage of QuaRot for network rotation invariance and introduces a lot of online computation in the design of the inference system.
>
> After you understand the essential differences between QuaRot and DuQuant, we'll answer your question below about why we didn't compare with DuQuant.

---

> ### Author Response · Authors · 2024-11-14
> **Rebuttal for Reviewer 6qo1-Part1**
>
> **Q1**: The reviewer think our observation appears trivial and unworthy of further investigation.
>
> **A1**: **We are completely unable to understand why the reviewer stated this opinion**. From our perspective, understanding why QuaRot's proposed approach to rotation improves the model's performance at quantization, especially under W4A4 quantization, can provide very valuable guidance for LLM's lower bit networks.
>
> Our work starts from the rotational invariance mentioned in QuaRot, and from the perspective of quantization error, we have carried out a reasonable and rigorous visual analysis and experimental evidence, and obtained that Rh can significantly improve the quantization of LLM under W4A4 compared to RO rotation simply because RH can better achieve better quantization under the massive activation of LLM. error, for the vast majority of tokens (even more than 99%) the two are unable to produce a significant difference, which we believe is extremely valuable. Our work helps the community to further understand the results of evaluating LLM quantization based on rotational invariance, which helps the community to further promote the research on LLM based on rotational invariance.
>
> At the same time, our method is identical to QuaRot in terms of inference speed because our method uses the same rotation matrix $R_1$ throughout the network as QuaRot, which allows $R_1$ to still be reparameterized into the weights of the network. Based on the fact that QuaRot is already supported by llama.cpp, we believe that our method only needs to provide the optimized rotation matrix (which often takes very little time to complete), and that our method can be seamlessly plugged into llama.cpp, as well as into any inference engine, as our method does not change the structure of the network and does not introduce any extra overhead (but DuQuant changes the structure and increases the floating point overhead of the network).
>
> **Q2**: The novelty and effectiveness of the method seem limited.
>
> **A2**: **With all due respect, I think we are working very efficiently and innovatively.**
>
> The overall lineage of our work is as follows: we have found that the reason for the huge performance difference between different rotation matrices (RO and RH) is the difference in the effect of Massive Activation, and have illustrated this through rigorous experiments, which are of great significance and which have helped to understand the specific behavior of rotational invariance in LLM quantization. To address this problem, we rationally abstract it to a long-tailed distribution and use the orthogonal Pluck transform to rationally solve the problem. Our approach is extremely rigorous and elegant in terms of experimental design and solution. At the same time, our method is efficient in the quantization process, and our method improves the modeling effect of QuaRot without introducing any additional computational overhead in inference, and it can be seamlessly inserted into llama.cpp, which is of significant application value.
>
> **Q3**: The method does not include evaluations related to actual memory reduction or speedup.
>
> **A3**: Our approach involves no modifications to the network structure, no kernel involvement, and DFRot and QuaRot have exactly the same overhead for inference, so we do not consider this to be necessary for the paper.
>
> **Q4**: Is the rotation matrix a Diagnol Block matrix?
>
> **A4**: No, $R_1$ is a dense $D\times D$ matrix, this is because as we know from the previous section, $R_1$ does not introduce any additional inference overhead, and we follow QuaRot's design of optimizing using Dense's rotation matrices.
>
> By the way, **I would again like the reviewers to carefully understand the differences between QuaRot and DuQuant as explained above**, and to understand that they are not at all the same type of work, and this will help you to understand why we are using Dense's rotation matrices instead of the block diagonal matrices mentioned in DuQuant.

---

> ### Author Response · Authors · 2024-11-14
> **Rebuttal for Reviewer 6qo1-Part2**
>
> **Q5**: How similar is the final matrix obtained by your method and the Hadamard matrix used for initialization?
>
> **A5**: They're completely different. The rotation matrix R1 optimized by our method is more like a “random” orthogonal matrix because our rotation matrix is optimized for the features, adapts to the quantization error of the features, and achieves a better elimination of the overall quantization error. (Incidentally, if you look at SpinQuant's open source optimized rotation matrix, you will see that there is almost no significant difference between the SpinQuant trained matrix and the initial Hadamard matrix.)
>
> **Q6**: Do different layers or projections share a common rotation matrix?
>
> **A6**: We used the same $R_1$ for the different layers, which is the starting point of our work, as it is the only way we can reason without introducing any additional overhead, i.e., following the rotational invariance of the network as stated in QuaRot.
>
> **Q7**: Compare with SpinQuant.
>
> **A7**: **We have shown a comparison between DFRot and SpinQuant in the appendix and compared the differences in depth**. We would like the reviewer to understand that SpinQuant, which is based on the WikiText-2 training data and PPL for the rotation matrix, may lead to ''overfitting'' to the WikiText-2 data, so that although it can improve significantly on the WikiText-2 PPL, it does not produce significant results on the zero-shot task. In other words, we believe that the performance improvement on WikiText-2 PPL does not strictly demonstrate the capability of the quantized LLM, and it is important to investigate better evaluation metrics of the quantized model for the development of LLM quantization field.
>
> **Q8**: On the visualization of quantization errors.
>
> **A8**: We will add a visualization of the quantization error in a later release. But before that, I would like reviewers to read the previous questions carefully, which will help you to further understand the differences between QuaRot and DuQuant, as well as the motivation and contribution of our work.
>
> Sincerely, the change in quantization error, while meaningful, is not a very fundamental difference. As in the case of cross-entropy loss when performing an image classification task, we can always tell that the loss goes down during the optimization process, but very little work has gone into showing the loss during the optimization process in papers.
>
> **Q9**: Large LLM (e.g., 70B).
>
> **A9**: We are very sorry that limited by computational resources, our lab is not equipped to explore larger models. However, we show further results of our method on QWen2-7B in the Appendix.
>
> Meanwhile, based on the fact that Massive Activation is also present in larger LLMs (e.g., 70B) as found in Massive Activation [1], it is entirely reasonable to infer that our method is equally applicable to larger models.
>
> [1] Sun, M., Chen, X., Kolter, J.Z. and Liu, Z., 2024. Massive activations in large language models. arXiv preprint arXiv:2402.17762.

---

> ### Comment · Reviewer_6qo1 · 2024-11-14
> **Response to Rebuttal: "Why DFRot Didn’t Compare It to DuQuant: They Are Completely Different"**
>
> Thank you for the authors' response. I would like to kindly address several claims that I find confusing.
> 1. "The introduction of $R_1$ doesn't introduce any additional computational overhead." Is this statement accurate?
>
> 2. "This leads to a large number of floating-point matrix multiplications, ..., which is not fully W4A4." From my perspective, QuaRot also introduces extra floating-point matrix multiplications. For instance, when multiplying K by V, we need to multiply K/V by a Hadamard matrix first. Is this operation fully W4A4? Additionally, if I'm correct, the QuaRot setting is K4V4Q16. Does this qualify as fully W4A4?
>
> 3. Regarding the second point you mentioned, are you aiming to highlight the limitations of your method?
>
> 4. Your response appears to miss the mark, as it should provide a comparison between your method and other methods, rather than between QuaRot and other methods.
>
> 5. Finally, while DuQuant was not compared, what about SpinQuant? I believe SpinQuant is more similar to your approach than DuQuant.

---

> ### Author Response · Authors · 2024-11-14
> **Response to additional question**
>
> Thank you for the authors' response. I would like to kindly address several claims that I find confusing.
>
> **Q1**: The introduction of doesn't introduce any additional computational overhead.
>
> **A1**: This is strictly accurate for $R_1$. Thanks to the property $RMSNorm(XR_1) = RMSNorm(X)R_1$ (which can be obtained from QuaRot, SliceGPT), $R_1$ can be merged by reparameterization into weight, changing ($W_{embedding}$, $W_{lm head}$, $W_q$, $W_k$, $W_v$, $W_o$,
> $W_{up}$,$W_{gate}$, $W_{down}$) into ($W_{embedding}R_1$, $R_1^TW_{lm head}$, $R_1^TW_q$, $R_1^TW_k$, $R_1^TW_v$, $W_oR_1$,
> $R_1^TW_{up}$,$R_1^TW_{gate}$, $W_{down}R_1$).
>
>
> **Q2**: Is this operation fully W4A4? Additionally, if I'm correct, the QuaRot setting is K4V4Q16. Does this qualify as fully W4A4?
>
> **A2**: First of all, you should understand that W4A4 as it is considered in LLM does not include KVCache compression (it full name should be W4A4KV16), and our task in this paper does not make any comparisons to KVCache.
>
> Second, the purpose of our paper is to provide further insights into the rotational invariance proposed by QuaRot, to understand the differences in the effects caused by rotational invariance for different types of tokens (normal tokens and tokens with massive activation), and to further draw out the contributions of our paper.
>
> Finally, if you read DuQuant's code carefully, you will realize that DuQuant did not investigate the compression of KVCache, they use KVCache with 16 bits, which is why we do not compare with them.
>
> **Q3**: Are you aiming to highlight the limitations of your method?
>
> **A3**: I agree that there are limitations in our work, for example, we only targeted $R_1$ in our optimization process, this is because
>
> 1. the optimization of $R_1$ is the most challenging of the rotational invariants because $R_1$ runs through the entire network, the rest of the optimization of $R_2$, $R_3$, and $R_4$ we can optimize by Block by Block; there is no optimization of $R_2$, $R_3$, and $R_4$ in our paper This is our future work, and we believe that the optimization of $R_1$ in our paper is sufficient to support the whole task;
> 2. The massive activation mentioned in our paper only occurs in the network locations that $R_1$ can correspond to, and other locations will not be characterized by this massive activation, which means that this kind of token will not be encountered. the problems mentioned in the paper;
> 3. We have made some simplifications to our work in optimizing $R_1$, i.e., obtaining the calibration set, optimizing $R_1$. In fact, from the description above, since $R_1$ affects the quantization results of the preorder network, a better optimization of $R_1$ would be to alternate obtaining the calibration set-optimizing $R_1$, but this would inevitably lead to an increase in the overhead of optimizing $R_1$. The starting point of our paper is to optimize $R_1$ with the smallest possible code, and the alternating optimization mentioned here is also our future work (in progress), which we are reasonably confident will yield better results.
>
> **Q4**: Your response appears to miss the mark, as it should provide a comparison between your method and other methods, rather than between QuaRot and other methods.
>
> **A4**: That's a good question.
>
> The reason about why we do not compare with previous methods is based on our consideration of the experiments demonstrated by SpinQuant[1] in W4A4.
>
> From the experiments we can recognize that previous scale-based (SmoothQuant, OS+, OminiQuant, AWQ, etc.) have invariably lost their competitiveness when the activation values are 4-bit, i.e., the performance of the model has completely collapsed. Currently, to the best of mt knowledge, at 4-bit per-token activation (no per-group quantization), **only methods based on rotational invariance are able to preserve the model's performance without a complete collapse**, even though a significant performance degradation still occurs compared to 8-bit activation values.
>
> At the same time, only half a year has passed since the proposal of work based on rotational invariance in the quantization process, and currently the only method that can provide a fair comparison based on rotational invariance (specifically referring to models like QuaRot) is QuaRot.
>
> These considerations combine to form the basis of our comparison with QuaRot alone in our paper.
>
> **Q5**: Compare with SpinQuant.
>
> **A5**: You can see it in Appendix as **Part-2 Q7** mentioned.

---

> > ### Comment · Reviewer_6qo1 · 2024-11-14
> > **Response to Rebuttal**
> >
> > I would like to kindly remind the authors to remain calm during the discussion period. If my concerns are addressed, I would happily raise my overall rating. However, there are still some issues that need attention.
> >
> > 1. Rotation Matrix Selection: Firstly, the authors did not provide descriptions for RO and RH. From my perspective, RH resembles a typical Hadamard matrix, while RO is a randomly initialized orthogonal matrix. The Hadamard matrix has favorable properties that allow it to evenly distribute any outliers (with elements ±1/(2^n)), including massive outliers, across other channels. In contrast, RO’s randomness does not guarantee the elimination of massive outliers. This seems evident to me, so I am unsure why the authors responded with, "We are completely unable to understand why the reviewer stated this opinion." Authors are responsible for clarifying all terms used in the paper; leaving reviewers to guess may have a negative impact on the paper’s reception. Furthermore, the authors have devoted excessive words to this minor point, making the overall structure difficult to follow. Additionally, no theoretical analysis is provided to substantiate this point. Overall, I view this as a weakness.
> > 2. Baseline Comparison: i) I suggested two baselines for comparison: DuQuant and SpinQuant. I included DuQuant because it also targets massive outlier elimination. Although DuQuant seems to have a slightly different KV setting as you mentioned, it still quantizes KV to INT4, so I don’t expect the results to differ significantly. Furthermore, both DuQuant and QuaRot involve computational overhead, with little difference as reported in the original paper. Therefore, I see DuQuant as comparable to DFRot. **However, it seems the authors are reluctant to make a comparison with DuQuant. In that case, let’s set it aside, but there is still the question: what about SpinQuant?**
> > ii) I have reviewed Appendix C multiple times. The main takeaway appears to be that DFRot is faster and more efficient than SpinQuant, but what about the performance comparison? The authors attribute DFRot’s inability to surpass SpinQuant on the Wiki PPL results to SpinQuant’s tendency to overfit on this dataset. However, this explanation is not entirely convincing. If PPL metrics are unreliable and potentially misleading, why are they used throughout the paper? In my view, if the authors want to argue an overfitting issue, they should experiment with different calibration datasets (e.g., C4) and test on WikiText PPL rather than making unsupported claims. There are also other benchmarks that could be used (e.g., MTBench, LongBench). Additionally, comparisons based on just one model, LLaMA 2-7B (Tables 4 and 5), are insufficient, especially given the small margin between SpinQuant and DFRot. Considering the authors' limited computational resources, additional comparisons on other small models could strengthen the results. P.S. there is a typo in Table 4: OFMAF -> DFRot.
> > 3. Visualization of Quantization Error: The authors' response to Q8 mentions adding a visualization of the quantization error, which readers would find valuable. A comparison with other baselines would make the proposed method’s effectiveness even more apparent.

---

> ### Author Response · Authors · 2024-11-15
>
> **Q1**: Some concern for $RO$ and $RH$.
>
> **A1**: I will politely admit to the reviewer that we do have weaknesses in the description of RO and RH, and we would be happy to make changes in the content to further ensure that the difference between RO and RH is properly understood.
>
> $\text{RH}$: Given a Hadamard matrix H, a random binarization dignal matrix $A_{ii}$ in $\{+1,-1\}$, $M=HA$,so $({M})({M})^T=HAA^TH^T=I$,where $I$ is a IdentityMatrix.
>
> $\text{RO}$: $\text{RO}$ is obtained from QR decomposition, where $Q$ is orthogonal.
>
>
>
>
>
> **Q2**: In contrast, RO’s randomness does not guarantee the elimination of massive outliers.
>
> **A2**: **From the reviewer's response, I reconfirm that you did not understand our argument：**
>
>
> ```text
> The Hadamard matrix has favorable properties that allow it to evenly distribute any outliers (with elements ±1/(2^n)), including massive outliers, across other channels. In contrast, RO’s randomness does not guarantee the elimination of massive outliers.
> ```
> **This is just a phenomenon, not our conclusion,** and it looks like the reviewer did not carefully read Section 3.2 what we illustrated by visualization (Figure 2 to Figure 4) and experiments in Table 1.
>
> I very sincerely hope that the reviewers will carefully read this next paragraph (our conclusion) and carefully read Section 3.2, which is the most fundamental motivation of our paper.
>
> ```
> We found that the performance difference between RH and RO almost completely disappears by just keeping the token with massive activation as FP16.
>
> Our experiments reveal that the essential reason why RH can outperform RO is because RH can exhibit smaller quantization errors on tokens with massive activation.
>
> e.g. RH = RO + Tokens with massive actvation
> ```
>
> **Wouldn't it be interesting to see how this 0.1% token could cause an essential performance difference?**
>
> As it helps us understand why RH can significantly RO, using this understanding to improve the method (**Reviewer MtcK**). It also echoes Massive Activation [1], and justifies Eq.2 that we introduced in Section 3.3.
>
> **Q3**: Some concern about PPL.
>
> **A3**: 1. W4A4KV4 SpinQuant RTN, we train and test on different datasets with SpinQuant:
>
> | Model | LLaMA2-7B | LLaMA2-7B | LLaMA2-7B | LLaMA2-7B |
> |:---:|:---:|:---:|:---:|:---:|
> | Train | WikiText-2 | WikiText-2 | C4 | C4 |
> | Test | WikiText-2 | C4 | WikiText-2 | C4 |
> | PPL | 6.20 | 8.38 | 6.57 | 8.13 |
>
> | Model | W-A-KV | Train | PQ | WG | HS | A-e | A-c | LA | Avg. |
> |---|:---:|:---:|:---:|:-:|:---:|:---:|:---:|:---:|:---:|
> | LLaMA2-7B | 4-4-4 | Wiki | 76.66 | 65.98 | 72.78 | 70.92 | 42.06 | 70.12 | 66.42 |
> | LLaMA2-7B | 4-4-4 | C4 | 75.28 | 66.12 | 71.82 | 71.28 | 42.88 | 70.31 | 66.28 |
>
> From the experiments we can draw two conclusions:
> - For model quantization, PPL on the target dataset highly depends on the calibration dataset;
> - Strictly speaking, the PPL does not accurately evaluate the model's zero-shot capability: in the case of using the WikiText-2 and C4 datasets respectively, although the difference in the PPL of LLaMA2-7B on WikiText-2 is significant, the zero-shot capability of the two does not demonstrate a significant difference.
>
> 2. From the comparisons in Tables 4 and Table 5, we can see that although SpinQuant's PPL on wikitext is significantly better than DFRot, these two do not demonstrate a significant difference on the zero-shot tasks, and even DFRot outperforms SpinQuant on some of the zero-shot tasks while its PPL is much lower than that of SpinQuant (LLaMA2-7B W4A4KV16).
>
> 3. The PPL of the training of the LLM does not represent the final capability of the model: i.e., we can only see a decrease in the PPL during the training of the model, but we cannot evaluate the final capability of the LLM during the training.
>
> Sincerely, this is an open topic, and we hope that in the future there will be better ways of evaluating the quantitative quality of models than relying solely on calibrating and testing PPL on specific datasets, which in fact is not as effective as we would like it to be.
>
> **Q4**: There is a typo in Table 4: OFMAF -> DFRot and visualize Quantization Error:
>
> **A4**: Many thanks to the reviewers for spotting this typo, we will fix it, and in the meantime we are visualizing the quantification error, which we will update in the appendix as soon as possible (Please see Figure 36 to Figure 38).
>
> [1] Massive activations in large language models.
>
> [2] Can Perplexity Reflect Large Language Model's Ability in Long Text Understanding?.
>
> **Q6**: Additional comparisons on other small models could strengthen the results.
>
> **A6**: We are doing our best to conduct some relevant experiments, and for us, A100 is still very expensive and difficult to obtain, which is why we did the PTQ.
>
> **Until then, I still sincerely hope that the reviewers understand the conclusions of our paper(Q2 & A2), and if you are still unable to understand the core contribution of our work, you are welcome to continue to reply.**

---

> ### Author Response · Authors · 2024-11-15
> **Re-emphasize the difference between QuaRot and DuQuant**
>
> Again, I strongly recommend the reviewer to look at the new Figure 20 and Figure 21 added to the paper to make it clear that QuaRot and DuQuant are fundamentally quite different. the rotation matrix $R_1$ of DuQuant introduces a significant amount of on-line computation, but $R_1$ in QuaRot can be directly merged into the network weights. Only QuaRot can be called a rotational invariant of the network.
>
> Based on the rotational invariance, we re-emphasize that a comparison between DuQuant and QuaRot is completely inappropriate
> even though both study massive activation.
>
> **In fact, if you look at it just from the perspective of quantization error, tokens with massive activation are instead the most easy to quantize:**
> ```
> Does the reviewer notice from Fig. 2 that these tokens' quantization error with massive activation instead exhibit the smallest quantization error after RH, and it is the ordinary token that has a larger quantization error.
>
> Our Section3.2 is precisely to show that these tokens with minimal quantization errors have the property of severely affecting the model's effectiveness.
> ```

---

### Meta-Review · Area_Chair_cmeD · 2024-12-25

**Metareview:**

The submission studies the effectiveness of rotations in mitigating quantization effects in large language models. The paper is motivated by the observation that randomized Hadamard rotations outperform random orthogonal rotations for inputs with large activations. It then introduces an optimization frame work which seeks a rotation which simultaneously eliminates outlying features and large activations. The paper proposes an alternating directions method for this problem, which alternates between assigning quantization centers (similar to K means) and solving for the optimal rotation matrix (orthogonal procrustes). This optimization uses a weighted loss which more heavily emphasizes tokens with large activations.

The paper addresses a timely topic: quantization in large language models. Reviewers positively evaluated the paper’s efforts to identify the reason which random Hadamard outperforms random orthogonal, and found the paper’s proposals well grounded in experiments. The main issues raised on in the review concerned the relationship between the proposed method and existing works that learn a rotation matrix for quantization. Reviewers questioned both the conceptual connections & distinctions between these methodologies and how their performance compared to the proposed method.

The main strength of the paper is its experimental investigation of (and explanation for) the good performance of randomized Hadamard with heavy (4 bit) quantization. Based on this investigation, the paper proposes a new method for generating rotations for quantization. The paper could do a better job of differentiating this method from other works for rotation learning, both conceptually and experimentally. In particular, after interaction with the authors, reviewers retained concerns about a number of aspects of the experimental comparison, including the performance comparison with SpinQuant, and experiments on models of varying size.

**Additional Comments On Reviewer Discussion:**

As described above, the paper generated extensive discussion between authors and reviewers. Most of the discussion centered around the following issues:
- 1. The novelty and interest of the paper’s explanation for the good performance of random Hadamard compared to random orthogonal
- 2. The position of the proposed method in the literature on rotations for quantization, in particular its relationship to methods such as QuaRot, SpinQuant, Quip and DuQuant
- 3. The strength and implications of the experimental comparison.

The authors response convincingly addresses issue 1. Although a number of reasons for the good relative performance of random Hadamard can be hypothesized, the paper performs experiments which zero in on tokens with large activations as the source of the performance difference. Indeed, a number of reviewers praise this contribution.

The discussion also effectively addressed a number of factual question regarding the submission including the definition and relationship between RH and RO, and differences in setting between this work and other recent works on learning rotations for quantization.

At the same time, after considering author responses, reviewers retained a number of concerns about the paper’s experimental comparisons.

---

### Decision · Program_Chairs · 2025-01-22

Reject